# Sensitivity and Resistance of Oncogenic RAS-Driven Tumors to Dual MEK and ERK Inhibition

**DOI:** 10.3390/cancers13081852

**Published:** 2021-04-13

**Authors:** Antonella Catalano, Mojca Adlesic, Thorsten Kaltenbacher, Rhena F. U. Klar, Joachim Albers, Philipp Seidel, Laura P. Brandt, Tomas Hejhal, Philipp Busenhart, Niklas Röhner, Kyra Zodel, Kornelia Fritsch, Peter J. Wild, Justus Duyster, Ralph Fritsch, Tilman Brummer, Ian J. Frew

**Affiliations:** 1Department of Internal Medicine I, Hematology, Oncology and Stem Cell Transplantation, Medical Center-University of Freiburg, Faculty of Medicine, University of Freiburg, 79106 Freiburg, Germany; antonella.catalano@uniklinik-freiburg.de (A.C.); mojca.adlesic@uniklinik-freiburg.de (M.A.); rhena.klar@uniklinik-freiburg.de (R.F.U.K.); philipp.seidel1@web.de (P.S.); roehner.niklas@gmail.com (N.R.); kyra.zodel@uniklinik-freiburg.de (K.Z.); kornelia.fritsch@uniklinik-freiburg.de (K.F.); justus.duyster@uniklinik-freiburg.de (J.D.); ralph.fritsch@usz.ch (R.F.); 2Institute of Physiology, University of Zurich, 8057 Zurich, Switzerland; joachim.albers@merckgroup.com (J.A.); laura.brandt@dkf.unibe.ch (L.P.B.); tomas.hejhal@me.com (T.H.); Philipp.busenhart@usz.ch (P.B.); 3Zurich Center for Integrative Human Physiology, University of Zurich, 8006 Zurich, Switzerland; 4Signaling Research Centre BIOSS, University of Freiburg, 79104 Freiburg, Germany; tilman.brummer@mol-med.uni-freiburg.de; 5Institute of Molecular Medicine and Cell Research (IMMZ), Faculty of Medicine, University of Freiburg, 79104 Freiburg, Germany; thorsten.kaltenbacher@tum.de; 6Spemann Graduate School of Biology and Medicine, University of Freiburg, 79104 Freiburg, Germany; 7Department of Pathology and Molecular Pathology, University Hospital Zurich, 8006 Zurich, Switzerland; Peter.Wild@kgu.de; 8Comprehensive Cancer Center Freiburg (CCCF), Medical Center–University of Freiburg, Faculty of Medicine, University of Freiburg, 79106 Freiburg, Germany; 9German Cancer Consortium (DKTK), Partner Site Freiburg and German Cancer Research Center (DKFZ), 69120 Heidelberg, Germany; 10Department of Hematology and Medical Oncology, University Hospital of Zurich, 8006 Zurich, Switzerland

**Keywords:** oncogenic RAS, MEK inhibitor, ERK inhibitor, drug resistance, mouse tumor model, metastasis, undifferentiated pleomorphic sarcoma, pancreatic ductal adenocarcinoma

## Abstract

**Simple Summary:**

Mutations in *RAS*-family genes frequently cause different types of human cancers. Inhibitors of the MEK (mitogen-activated protein kinase) and ERK (extracellular signal-regulated kinase) protein kinases that function downstream of RAS proteins have shown some clinical benefits when used for the treatment of these cancers, but drug resistance frequently emerges. Here we show that combined treatment with MEK and ERK inhibitors blocks the emergence of resistance to either drug alone. However, if cancer cells have already developed resistance to MEK inhibitors or to ERK inhibitors, the combined therapy is frequently ineffective. These findings imply that these inhibitors should be used together for cancer therapy. We also show that drug resistance involves complex patterns of rewiring of cellular kinase signaling networks that do not overlap between each different cancer cell line. Nonetheless, we show that MAP4K4 is required for efficient cell proliferation in several different MEK/ERK inhibitor resistant cancer cell lines, uncovering a potential new therapeutic target.

**Abstract:**

Oncogenic mutations in *RAS* family genes arise frequently in metastatic human cancers. Here we developed new mouse and cellular models of oncogenic Hras^G12V^-driven undifferentiated pleomorphic sarcoma metastasis and of Kras^G12D^-driven pancreatic ductal adenocarcinoma metastasis. Through analyses of these cells and of human oncogenic KRAS-, NRAS- and BRAF-driven cancer cell lines we identified that resistance to single MEK inhibitor and ERK inhibitor treatments arise rapidly but combination therapy completely blocks the emergence of resistance. The prior evolution of resistance to either single agent frequently leads to resistance to dual treatment. Dual MEK inhibitor plus ERK inhibitor therapy shows anti-tumor efficacy in an *Hras^G12^*^V^-driven autochthonous sarcoma model but features of drug resistance in vivo were also evident. Array-based kinome activity profiling revealed an absence of common patterns of signaling rewiring in single or double MEK and ERK inhibitor resistant cells, showing that the development of resistance to downstream signaling inhibition in oncogenic RAS-driven tumors represents a heterogeneous process. Nonetheless, in some single and double MEK and ERK inhibitor resistant cell lines we identified newly acquired drug sensitivities. These may represent additional therapeutic targets in oncogenic RAS-driven tumors and provide general proof-of-principle that therapeutic vulnerabilities of drug resistant cells can be identified.

## 1. Introduction

The *KRAS*, *NRAS* and *HRAS* oncogenes are mutated in approximately 30% of all human cancers. KRAS mutations comprise roughly 85% of these cases, whereas NRAS (12%) and HRAS (3%) are less frequently mutated [1]. Tumor types that are commonly driven by *RAS* family gene mutations include pancreatic ductal adenocarcinoma (PDAC), colorectal carcinoma (CRC), non-small cell lung cancer (NSCLC), embryonal rhabdomyosarcoma (RMS) and undifferentiated pleomorphic sarcoma (UPS). Oncogenic RAS mutations typically affect amino acids that are critical for normal RAS regulation and function, namely the GDP (guanosine diphosphate)/GTP (guanosine triphosphate) molecular switch, which results in a constitutively active GTP-bound protein. Direct pharmacological inhibition of RAS proteins by way of attempts to generate GTP-competitive inhibitors has been complicated by the lack of drug-binding pockets outside of the nucleotide-binding pocket and by the picomolar binding affinity of GTP for RAS [2,3,4]. Nevertheless, several KRAS^G12C^-specific inhibitors, including AMG-150 and MRTX849, have been recently developed and are currently under investigation in Phase I/II studies [5,6,7]. Unfortunately, these inhibitors are not able to efficiently target other RAS mutants, narrowing their clinical utility. Targeting oncogenic RAS signaling, by altering membrane association, utilizing synthetic lethal effects, metabolism, or effector signaling, have also had limited success [8,9,10,11,12,13,14,15].

Several promising agents that target downstream RAF-MEK-ERK signaling pathways have been developed and are in pre-clinical and clinical testing [8,16,17]. Since it was discovered that inhibition of BRAF in RAS-driven tumors leads to paradoxical ERK activation [9,18], MEK and ERK inhibitors represent more promising candidates to interfere with signal transduction by oncogenic RAS. Allosteric non-ATP competitive inhibitors of MEK act by blocking ERK phosphorylation by MEK [17]. Tumor cells however frequently rapidly develop resistance to these agents, often due to ERK reactivation. Second generation MEK inhibitors belong to the so-called dual-mechanism inhibitor or “feedback busters” category that bind to and inhibit the intrinsic kinase activity as well as induce conformational change upon binding and prevent the phosphorylation of the target kinase itself [17,18,19]. This mechanism of action aims to reduce their vulnerability to the loss of ERK-dependent negative feedback loops [8,9]. Two novel dual mechanism MEK inhibitors, GDC-0623 and RO5126766, have successfully completed phase I clinical trials. ERK inhibitors have also recently been developed. One of the most promising current ERK inhibitors is the pre-clinical tool compound SCH772984, and its orally available analogue MK-8353, which has recently successfully completed a phase I study [20]. This drug also has a dual mechanism of action, causing concomitant allosteric inhibition of MEK1/2 binding and ERK1/2 phosphorylation, as well as the ATP-competitive inhibition of ERK1/2 phosphorylation of its substrates [20,21].

Cell culture studies and human clinical studies have revealed that the development of resistance to downstream signaling inhibitors in oncogenic RAS- and RAF-driven cancers occurs frequently. These resistance mechanisms arise as a consequence of the complex and intertwined signaling networks governed by RAS and the existence of multiple feedback mechanisms that in physiological conditions assure accurately timed pathway activation and duration [16,22]. Several studies have shown that single inhibition of either RAF or MEK alone is not sufficient to achieve a prolonged inhibition of the RAS pathway [18,23,24,25,26,27,28]. While there is not yet enough clinical data about ERK inhibition alone, resistance mechanisms have been predicted to arise [29,30]. Many resistance mechanisms arising after single RAF or MEK inhibition have been shown to be driven by ERK reactivation, thereby suggesting that concurrent inhibition of the pathway at multiple levels including ERK may induce a more effective tumor growth arrest [22,31]. In order to test whether resistance to these agents can be overcome by ERK inhibition, several in vitro studies were performed and treatment of tumor cell lines resistant to RAF and/or MEK inhibitors showed sensitivity to ERK inhibitors (SCH772984 or VTX-11e) [21,25]. Subsequent studies demonstrated that so-called vertical pathway inhibition of the RAF-MEK-ERK cascade could be achieved by concomitant administration of agents targeting different levels of the signaling pathway. Experiments performed on BRAF-mutant tumor cell lines and their xenografts showed that sequential monotherapy is ineffective because it leads to the parallel evolution of resistant BRAF-amplified clones [32]. However, concurrent inhibition of the RAF-MEK-ERK kinases using an intermittent treatment schedule of 3 days on-4 days off to limit toxicity and efficient inhibition of tumor growth in PDX (patient-derived xenograft) models [32]. Similarly, in RAS-mutant cancers, inhibition of MEK or ERK alone is not sufficient to suppress tumor growth but combined inhibition of the two kinases results in a deeper and more durable pathway suppression and tumor growth control in multiple xenograft models [33]. In *Kras*-driven genetically engineered mouse models of NSCLC (non-small-cell lung carcinoma) and PDAC (pancreatic ductal adenocarcinoma), dual MEK and ERK inhibition reduced tumor growth and increased progression-free survival, but it is important to note that the animals still died of the tumors, indicating that resistance to dual therapy arises in vivo and that this regime is unlikely to be curative in the clinic [33]. A more detailed understanding of mechanisms that underlie resistance to these agents is needed.

In this study we further investigate the effects of single and dual MEK and ERK inhibition in multiple oncogenic RAS-driven mouse and human cellular models in the context of therapeutic sensitivity and resistance, in order to gain insight into the factors that may limit this therapeutic strategy. In all cell models we show that resistance to single agent MEK inhibitor (GDC-0623) and ERK inhibitor (SCH772984) treatments arise rapidly but that combination therapy completely blocks the emergence of resistance. However, the prior evolution of resistance to either single agent frequently leads to subsequent resistance to dual treatment, with implications for clinical treatment scheduling. Kinome profiling of multiple resistant cell lines revealed an absence of common patterns of signaling rewiring, implying that the development of resistance to single and dual MEK and ERK inhibition in RAS-driven tumors represents a highly heterogeneous process. Nonetheless, in some cell lines, the targeted inhibition of kinases that are upregulated in MEK inhibitor or MEK inhibitor plus ERK inhibitor resistant cells specifically slowed cellular proliferation and greatly delayed resistance, providing proof-of-principle that newly acquired drug sensitivities that are specific for the resistant cells can be identified.

## 2. Materials and Methods

### 2.1. Generation of MuLE Vectors

MuLE vectors carrying shRNA-*Cdkn2a* + *Hras^G12V^*, shRNA-*Trp53* + *Hras^G12V^* and shRNA-*Trp53* + shRNA-*Pten* + *Hras^G12V^*, as well as empty entry vectors and all destination vectors used in this study were previously described [34]. Ecotropic lentiviral vectors were produced by using calcium phosphate-mediated transfection of sub-confluent HEK293T cells and the viral preparation was concentrated as described previously [34].

### 2.2. Mouse Strains

SCID/Beige mutant mice (C.B-17/CrHsd-PrkdcScidLystbg-J and CB17.Cg-*PrkdcscidLystbg-J*/Crl) were obtained from Envigo and Charles River Laboratories. The *LSL-Kras^G12D/+^* and *LSL-Trp53^R172H/+^* knock-in alleles as well as *Pdx-1::Cre* transgenic mice have been described previously [35] and were maintained on a C57BL/6N background. Animals were kept at 21–23 °C with 45–60% humidity and a 12 h dark/12 h light cycle under specific pathogen-free conditions in the animal facility of the University Medical Center Freiburg according to institutional guidelines. Mice received standard diet and water *ad libitum*.

### 2.3. In Vivo Tumour Formation and Metastases Study

Thirty μL of concentrated ecotropic lentivirus (titer of 10^6^ TU/mL) were injected into the left gastrocnemius muscle of 18–21-day old mice. In vivo non-invasive luciferase imaging was used to follow tumor development over time as described [34]. Mice were sacrificed as soon as one of the following endpoint criteria were reached: bioluminescence signal intensity more than 10^9^ p/s (photons/second), weight loss more than 20% of original body weight, poor body condition as well as unresponsive behavior.

### 2.4. Allograft Studies

Single cell suspension was prepared with Accutase, 1 × 10^6^ cells were suspended in 50% Matrigel (BD, no.354230) and injected subcutaneously in the flank of anesthetized SCID/Beige mice. In vivo imaging [34], as well as measurement of tumor volume using a caliper were used to follow tumor development over time.

### 2.5. In Vivo Imaging Studies

In vivo bioluminescence imaging was performed using the IVIS Spectrum or IVIS Lumina III, Perkin Elmer, together with the Living Image software (version 4.4 or 3.2/4.5 respectively). Mice were anaesthetized using a vaporized isoflurane in O_2_ in an inhalation narcosis chamber. Mice were weighed and injected subcutaneously with 150 mg/kg D-luciferin using a 30G insulin syringe. Quantitative luciferase imaging was performed 12 min after injection when peak signal intensity is reached. For the quantification of the total radiant efficiency, a region of interest (ROI) was drawn around the tumor and the total radiant efficiency (Total flux (p/s)) was automatically detected.

### 2.6. Preclinical Therapeutic Study

Ecotropic lentivirus was intramuscularly injected and once the tumors reached a bioluminescence signal intensity between 5 × 10^8^ and 1 × 10^9^ p/s for the short-term treatment or between 5 × 10^7^ and 1 × 10^8^ p/s for the long-term treatment, 2 to 3 mice in each of the cohorts or 5 to 7 mice in each of the cohorts, respectively for the short and long-term treatments, were treated according to a “5 days on-2 days off” schedule [32]. Therapies were administered via oral gavage or intraperitoneal (ip) injections with selected inhibitors, either as single treatment or combined therapy, and their corresponding vehicle for the control groups. Inhibitors used in this study were the MEK inhibitor GDC-0623 (Selleckchem S7553) administered 40 mg/kg *per os* (gavage) once/day and the HCl-salt form of the ERK inhibitor SCH772984 (custom synthesized by Selleckchem, Houston, TX, USA) administered 50 mg/kg parentally (ip injection) twice/day. Vehicles were 0.1% Methylcellulose + 0.1% Tween 80% and 10% Hydroxypropyl beta-cyclodextrin (HPCD) in 0.9% NaCl respectively. Solutions of 5 mg/mL and 25 mg/mL respectively were prepared freshly every two/three days and stored at 4 °C. Mice were monitored daily with regards to their body weight and general health and tumor growth was assessed every 5–8 days by in vivo bioluminescence imaging [34]. Mice were sacrificed as soon as one of the following endpoint criteria was reached: bioluminescence signal intensity ≥10^9^ p/s (photons/second), weight loss ≥20% of original body weight, poor body condition as well as unresponsive behavior.

### 2.7. Cells

HEK293T and NIH3T3 cells were originally purchased from ATCC^®^ (CRL-3216^TM^ and CRL-1658^TM^ respectively). Mouse primary tumor (UPS Pr) and metastatic lesions-derived cells (UPS Li/Lu) were isolated from approximately 1/3 of a dissected primary tumor and from luciferase-positive areas (metastatic lesions) of liver and/or lung of SCID-Beige mice injected with *shRNA-Trp53* + *Hras^G12V^* lentiviral vector as described [34]. Mouse pancreatic tumor (KPC Pr Sol/Sof) and metastatic lesion-derived cells (KPC Li/Sp) were isolated from dissected tumor-bearing organs by digestion for 30 min at 37°C with 0.05% Trypsin/EDTA solution followed by direct plating in DMEM (Dulbecco’s modified Eagle medium) plus 10% FBS (fetal bovine serum). Human non-small cell lung carcinoma- metastatic site (lymph node)- cell line NCI-H1299 was provided by the BIOSS toolbox, the central repository of the Centre for Biological Signaling Studies, University of Freiburg, human colon carcinoma cell line RKO was a kind gift of Prof. Nils Blüthgen, Berlin and human colon carcinoma cell lines SW480 and HCT116 were obtained from ATCC^®^ (CCL-228^TM^ and CCL-247^TM^ respectively). Human rhabdomyosarcoma cell line RD was kindly provided by Prof. Dr. Beat Schäfer (Children’s University Hospital Zurich). Human pancreatic cell lines Capan-2 and HPAF-II were purchased from ATCC^®^ (HTB-80^TM^ and CRL-1997^TM^ respectively) and cultivated in McCoy’s 5a medium supplemented with 10% FCS and 1% Penicillin-Streptomycin and in Eagle’s Minimum Essential Medium supplemented with 10% FCS and 1% Penicillin-Streptomycin respectively. Primary patient-derived PDAC cell lines B23 and B40 were isolated as described [36] and cultivated in 2D in Human Complete Feeding Medium supplemented with Wnt3a-conditoned Medium. KRAS mutational state was identified as described [37]. All other cells used in this study were cultured in DMEM high glucose medium supplemented with 10% FCS and 1% Pen/Strep and maintained in a humidified 5% (*v*/*v*) CO_2_ and 20% O_2_ incubator at 37 °C. Cells were tested for mycoplasma via the Mycoplasmacheck service provided by Eurofins Genomics.

### 2.8. Scratch Assay

Scratch assay was performed by seeding 9 × 10^5^ cells per 6-well plate in triplicate. The day after, a first picture of the confluent plate was taken and a scratch with a 200 μL pipette tip was performed on the cell monolayer. Pictures of a defined/marked area were taken at the time of the scratch (t0), after 4 (t1), 8 (t2), 12 (t3) and 24 (t4) hours. Pictures were captured with Axiocam MRc5 camera connected to Axiovert 40 CRC microscope. Images were acquired with Axiovision LE software and were analyzed by using ImageJ software. Data were plotted as percentage of recovery relative to t0.

### 2.9. Transwell Migration Assay

Transwell migration assay was performed in 24-well plates (Corning 353504) with 8.0 μm-pore polyester membrane inserts (Corning 353097) following the Corning manufacturer’s instructions. Briefly, inserts were coated with 100 μL of 300 μg/mL Matrigel in coating buffer (0.01M Tris pH 8.0, 0.7% NaCl) and incubated for 2 h at 37 °C. Twenty five thousand primary tumor and metastatic lesion-derived cells were trypsinized and after inactivation of trypsin with DMEM + 10% FCS, the cells were resuspended in 500 μL of DMEM without serum. Cells were seeded inside the inserts and 750 μL chemoattractant (DMEM + 5% FCS) added to the bottom of each well. Invasion chambers were then incubated overnight in humidified 5% (*v*/*v*) CO_2_ and 20% O_2_ incubator at 37 °C. Inserts were removed from the chambers, washed twice in PBS and the cells were fixed in 2% PFA (paraformaldehyde) for 10 min at room temperature. After washing twice in PBS, cells were subsequently stained with DAPI (1:100) for 10 min at room temperature, followed by one more washing step. Membranes were then excised from the inserts, mounted on microscopy slides with Mowiol solution (Mowiol 4-88 Calbiochem 475904) and stored at −20 °C. All experiments were performed in triplicates. Ten representative pictures of each membrane were captured with Axiocam MRc5 camera connected to Zeiss Scope A1 microscope. Images were acquired with Axiovision LE software and cells counted with Cell Profiler software.

### 2.10. Proliferation and Viability Assay of Inhibitor-Treated Cells

Thirty thousand cells per well were seeded in 6-well plates in triplicates. Medium was changed 24 h after plating and 48 h after seeding (day 0) cells were treated with 1 μM of the inhibitor(s) or the equivalent DMSO (dimethyl sulphoxide) volume for three days. Cells were trypsinized and analyzed with CASY TT Cell Counter and Analyzer System (OLS OMNI Life Science 5651697) that automatically provided cell count and viability values.

### 2.11. Short-Term Drug Screening 

Cells were seeded at a density of 2000 cells per well in 96 well plates. Twenty hours after seeding the medium was changed and 48 h after seeding (day 0) the cells were treated with 1 μM of DMSO or inhibitor(s) for three days. After completion of the drug exposure, cells were analyzed using the sulforhodamine B (SRB) colorimetric assay as described [38]. The relative cell growth was calculated on the average OD of the sample versus/normalized to average OD at day 0 and OD of the vehicle (DMSO)-treated control. 

Inhibitors used in our study were: RO5126766 (CH5126766) (Selleckchem, Houston, TX, USA, S7170), GDC-0623 (Selleckchem, Houston, TX, USA, S7553), SCH772984 (Selleckchem, Houston, TX, USA, S7101). All inhibitors were dissolved in DMSO (D1435, Sigma Aldrich, St. Louis, MO, USA) and used at 1 μM final concentration. Final DMSO concentration was kept at 0.1% or below in control and inhibitor-treated cells for single treatments and between 0.2% and 0.3% in control and inhibitor-treated cells for double and triple treatments respectively. 

### 2.12. Drug Interaction Assay

Two thousand cells per well were seeded in 96 well plates. Twenty four hours after seeding, medium was changed and after another 24 h (day 0) cells were treated with different concentrations of the inhibitors for three days. Concentration range was chosen for each drug according to previous empirically-determined IC50 values and applying a dilution factor of 2. After completion of the drug exposure, cells were analyzed using the sulforhodamine B (SRB) colorimetric assay as described [38]. Absorbance values were used to determine the coefficient of drug interaction (CI) with the Anaconda^®^ 5.3_jupyter 5.6.0 software. The software calculates and plots drug-response and drug interaction curves by using the method of isoboles. Dose-response curves (ellipsoid) were obtained according to the formula b = B*(1 − (a/A)**n)**(1/n) where a and b are the doses of Drug A and Drug B respectively when the two are present together, A and B are the respective individual doses for that effect level (IC50) and n is curvature (*n* = 1 additive, *n* < 1 superadditive (also known as synergistic), *n* > 1 subadditive) [39]. 

### 2.13. Drug Resistance Assay

Long-term drug resistance was assessed by seeding in parallel 2.5 × 10^5^ and 5 × 10^5^ cells in 6-cm plates and 24 h after plating cells were treated with 1 μM of a particular inhibitor as single or double treatment. For the plates where the initial cell number was 2.5 × 10^5^, medium and inhibitor(s) were changed every 3 days for 14 days. Cells were fixed and stained with Crystal Violet solution, pictures were captured with EPSON PERFECTION V300 PHOTO scanner and images acquired with the paired software. taken. For the plates whose initial cell number was 5 × 10^5^, medium and inhibitor(s) were changed every 3 days until plates were fully confluent. At confluence plates were split 1:5 to one 6-cm-plate without inhibitor(s) (1/5), in order to test for possible drug addiction, a 10-cm-plate with inhibitor(s) (3/5) for expansion/freezing and one 6-cm-plate with inhibitor(s) (1/5) for future protein isolation. Days after resistance were counted from the splitting time until the plates reached confluence again. Inhibitors used were RO5126766 (CH5126766) (Selleckchem, Houston, TX, USA, S7170), GDC-0623 (Selleckchem, Houston, TX, USA, S7553), SCH772984 (Selleckchem, Houston, TX, USA, S7101), LY3214996 (Selleckchem, Houston, TX, USA, S8534). Final DMSO concentration was kept at 0.1% in control and inhibitor-treated cells for single treatments and 0.2% for double treatments.

### 2.14. Colony Formation Assay

Clonogenic assay was performed by seeding 500 cells per 6-well plate in triplicate. Medium containing 1 μM DMSO (dimethyl sulfoxide) or drug(s) was added 48 h after plating and refreshed every 3 days for 14 days. Medium was removed and after washing once with PBS, cells were fixed and stained in 0.3% Crystal Violet solution in 70% MetOH for 30–60 min at room temperature. Staining solution was removed, plates washed in running tap water and let air-dry overnight. Plates were stored at room temperature. Pictures were captured with EPSON PERFECTION V300 PHOTO scanner and images acquired with the paired software.

### 2.15. EGF Stimulation of KPC Cells

Confluent cells were stimulated by adding 10 ng/μL EGF (Epidermal Growth Factor) (Recombinant murine EGF, PeproTech, Rocky Hill, NJ, USA) for 5 min into the medium. After incubation time, medium was immediately removed and cells were washed with ice cold PBS. Total cell lysates were analyzed via Western Blotting.

### 2.16. Western Blotting

Cultured cells were lysed in RIPA (Radioimmunoprecipitation assay) buffer (50 mM Tris–HCl at pH 8.0, 150 mM sodium chloride, 1% (*v*/*v*) NP340, 0.5% (*w*/*v*) sodium deoxycholate, 0.1% (*w*/*v*) SDS (sodium dodecyl sulfate), 5 mM sodium fluoride, 1 mM sodium orthovanadate, 1mM PMSF (phenylmethylsulphonyl fluoride) and Protease Inhibitor Cocktail (1:100, Sigma Aldrich)) and muscle tissue protein extract was prepared as described [40] followed by homogenization in RIPA buffer. 50 mg of protein lysate were run on 10–12% polyacrylamide gel, transferred to nitrocellulose membrane and visualized by immunoblotting with the following primary antibodies: anti-DESMIN antibody (1:1000, Sigma-Aldrich, St. Louis, MO, USA D1033), anti-MYOD1 antibody (1:500, Dako, Santa Clara, CA, USA, M3512), anti-MYOGENIN antibody (1:1000, Dako, Santa Clara, CA, USA, M3559), anti-H-RAS antibody (1:1000, Santa Cruz Biotechnology, Dallas, TX, USA, sc520), anti-VINCULIN (EPR8185) antibody (1:10,000, Abcam, Cambridge, United Kingdom, ab129002), anti-β-actin antibody (1:2000, Sigma Aldrich, St. Louis, MO, USA, A2228), anti-phospho-p44/42 MAPK (ERK1/2) (Thr202/Tyr204) (1:1000, Cell Signaling, Danvers, MA, USA, 9101), anti-p44/42 MAPK (ERK1/2) (137F5) (1:1000, Cell Signaling, Danvers, MA, USA, 4695S), anti- phospho- FRA1 (Ser265) (D22B1) (1:1000, Cell Signaling, Danvers, MA, USA, 5841), anti-FRA1 (D80B4) (1:1000, Cell Signaling, Danvers, MA, USA, 5281), anti-BRAF/Raf-B (H-145) (1:2000, Santa Cruz Biotechnology, Dallas, TX, USA, sc-9002), anti-CRAF/Raf-1 (C-12) (1:750, Santa Cruz Biotechnology, Dallas, TX, USA, sc-133), anti-MEK 1/2 (1:1000, Cell Signaling, Danvers, MA, USA, 9122), anti-phospho-MEK1/2 (Ser217/221) (41G9) (1:1000, Cell Signaling, Danvers, MA, USA, 9154), anti-phospho-EGFR (Y1068) (D7A5) (1:1000, Cell Signaling, Danvers, MA, USA, 3777), anti-EGFR (D38B1) (1:1000, Cell Signaling, Danvers, MA, USA, 4267), anti-phospho-AKT (T308) (C31E5E) (1:1000, Cell Signaling, Danvers, MA, USA, 2965), anti-AKT (1:1000, Cell Signaling, Danvers, MA, USA, 9272), anti-GAPDH (1:500, Abcam, Cambridge, MA, USA, ab9484). Signal detection was carried on through the enhanced chemiluminescence (ECL) reaction and visualized using Fujifilm Luminescent Image Analyzer LAS4000 (GE Healthcare, Chicago, IL, USA), a Fusion Solo chemiluminescence reader (VILBER LOURMAT), or through a fluorescence-based reaction and visualized using the Odyssey^®^ CLx (LI-COR Biosciences, Lincoln, NE, USA).

### 2.17. PCR Analysis of Genetic Kras Recombination

Genomic DNA was extracted from confluent dishes after trypsinisation followed by digestion in DNA lysis buffer containing Proteinase K (AppliChem GmbH, Darmstadt, Germany). PCR was performed as described [41] and a detailed protocol can be found at https://jacks-lab.mit.edu/protocols/genotyping/kras_cond. Following oligonucleotides were used as primers for the PCR reaction: *Kras*-universal (5′–3′ CCTTTACAAGCGCACGCAGACTGTAGA), *Kras*-mutated (5′–3′ AGCTAGCCACCATGGCTTGAGTAAGTCTGCA), Krasrecomb I (5′-3′ GTCTTTCCCCAGCACAGTGC), Krasrecomb II (5′–3′ CTCTTGCCTACGCCACCAGCTC), Krascomb III (5′–3′ AGCTAGCCACCATGGCTTGAGTAAGTCTGCA). PCR products were resolved in 2% agarose gels. PCR products were further analyzed via sequencing after excision of bands under UV light and DNA gel extraction procedure. 20 μL of purified DNA were sent to GATC biotech in Konstanz. Sequencing results were analyzed by using Snapgene alignment function.

### 2.18. Flow Cytometry

In order to quantify the frequency of cancer stem cells in the isolated KPC (*K**ras^G12D^*-*Trp53^+/R172H^*
Cre driven pancreatic ductal adenocarcinoma model) cell lines, cells were stained for CD44 (BD Bioscience, Franklin Lakes, NJ, USA, 561859), CD24 (BD Bioscience, 553262) and CD133 (Anti-prominin) (Milteny Biotec, Bergisch Gladbach, Germany, 130-123-793). Thereafter, 1 × 10^6^ cells were counted, transferred into a flow cytometry tube, centrifuged, washed and resuspended in 50 μL of triple stain solution containing 1:200 CD44-PE, 1:100 CD24-FITC and 1:5 CD133-APC diluted in FACS buffer. The corresponding single stain controls and unstained controls were carried along throughout the experiment. Cells were incubated for 30 min in the dark on ice with the staining solution. Then 3 mL FACS buffer was added to each FACS tube and cells were centrifuged for 5 min at 1500 rpm at 4 °C. Supernatant was subsequently aspirated and cells were resuspended in 1 mL of FACS buffer. Finally, cells were analyzed with LSRII flow cytometer. Unstained control was used to set voltage of lasers and living gates and single stained controls were used for compensation of the samples. FlowJo V10 was used to analyze FACS data and to generate Dot plot images.

### 2.19. PamChip^®^ Cell-Based Kinase Assay

Kinase activity profiles were determined using the phosphotyrosine kinase (PTK) and serine-threonine kinase (STK) PamChip^®^ peptide microarrays on the PamStation12 (PamGene International BV) according to manufacturer’s instructions. Parental and resistant cell lines were lysed in M-PER^TM^ Mammalian Extraction Buffer (Thermo Fischer Scientific, Waltham, MA, USA, 78503) supplemented with Halt^TM^ Phosphatase Inhibitor Cocktail (100×) (Thermo Fischer Scientific Waltham, MA, USA, 78428) and Halt^TM^ Protease Inhibitor Cocktail (100×) (Thermo Fischer Scientific Waltham, MA, USA, 78425) and 1 μg or 5 μg of protein lysate for PTK or STK respectively was applied to individual PamChip^®^ 4 arrays. Each array contains either 196 tyrosine or 144 serine-threonine peptides immobilized on a porous ceramic membrane. The peptide sequences (13 amino acids long) harbor phosphorylation sites derived from literature or computational predictions and are correlated with one or more upstream kinases. Kinases present in the lysates will phosphorylate the peptide substrates, which are detected using fluorescently labelled antibodies. Parental and resistant cell lines lysates were incubated on the arrays in the presence of ATP to facilitate the phosphorylation of peptides by protein kinases in the lysates. The in vitro long-term effect of MEK and ERK inhibitors GDC-0623 and SCH772984 on kinase activity profiles of 9 independent cell lines was determined. Fluorescent signal intensities for each peptides were analyzed with Bionavigator63 software. Visual quality control was performed to exclude defective arrays from the analysis. Changes in kinase expression were indicated by mean kinase scores as Log2 Fold Change (LFC) of each resistant cell line compared to their parental counterpart. A Z score within each subset of resistant mean kinase scores was also calculated. Intersecting sets of commonly altered kinases were represented using an adapted code from the package UpSetR [42]. The original technique and the interactive visualization tool implementing the approach were described by [43].

### 2.20. Immunohistochemistry

Tumors were dissected, fixed in 10% formalin, paraffin-embedded and cut in 3–5 μm sections. Immunohistochemistry analysis was performed as described [44]. The following antibodies were used in this study: anti-H-RAS antibody (1:100, GeneTex, Irvine, CA, USA, GTX116041), anti-Pax7 antibody (1:200, DSHB, Iowa City, IA, USA, AB_528428), anti-MyoD1 antibody (1:100, Dako, Santa Clara, CA, USA, M3512), anti-Myogenin antibody (1:500, Dako, Santa Clara, CA, USA, M3559), anti-VIMENTIN antibody (1:500, Cell Signaling, Danvers, MA, USA, D21H3), anti-SMA (1:200, Abcam Cambridge, MA, USA, ab5694), anti-PAN-CYTOKERATIN (1:100, BMA Biomedicals AG, Augst, Switzerland TI302), anti-CD31 antibody (1:200, Abcam, Cambridge, MA, USA, ab28364), anti-HMB45 (1:200 Abcam, Cambridge, MA, USA, ab732), anti-phospho-p44/42 MAPK (ERK1/2) (Thr202/Tyr204) (1:200, Cell Signaling, Danvers, MA, USA, 9101), anti-phospho-FRA1 (Ser265) (1:200, Biorbyt, Cambridge, United Kingdom, orb606).

### 2.21. Statistics and Data Evaluation

Statistical significance of all experiments was determined via the GraphPad Prism 7 software and the appropriate statistical test was chosen among the built-in analyses according to the data distribution. Two-way ANOVA (analysis of variance) method or 2-tailed Student’s or Welch’s *t*-tests with a *p* value of less than 0.05 considered to be statistically significant were used.

### 2.22. Study Approval

Mouse experiments were approved by the Veterinary Office of the Canton of Zurich under the license 137/2013 and the Center for Experimental Models and Transgenic Services (CEMT-FR) of the city of Freiburg under the licenses G13/050 and G17/081. Human pancreatic cancer cell lines were generated from surgically resected tumors with patient’s permission in the context of study IRB126/17 of the Medical Center–University of Freiburg.

## 3. Results

### 3.1. Development of an Hras^G12V^-Driven Mouse Model of Metastatic Undifferentiated Pleomorphic Sarcoma

We previously described the generation of three autochthonous models of sarcoma via intramuscular injection of concentrated ecotropic MuLE lentiviruses expressing either shRNA-*Cdkn2a* + *Hras^G12V^*, shRNA-*Trp53* + *Hras^G12V^* or shRNA-*Trp53* + shRNA-*Pten* + *Hras^G12V^* in SCID/Beige mice [34]. The lentiviral constructs also carried a luciferase expression cassette permitting bioluminescence imaging (BLI) in order to follow tumor growth and possible metastatic spread [34]. Our initial histological analysis of the tumors revealed that they were undifferentiated sarcomas with pleomorphic and rhabdoid features, with rare isolated malignant cells showing rudimentary sarcomere formation, thereby suggesting myogenic differentiation in some cells [34]. To better define the pathological description of these tumors we carried out immunohistochemical stainings using a panel of muscular markers that reflect different stages of the maturation of muscle cells, as well as for markers of the most common vascular, epithelioid and fibroid tumors (Appendix A). These analyses revealed that the tumors lacked expression of any markers of muscle differentiation (PAX7, MYOD, MYOGENIN) but were positive for VIMENTIN, while some spindled and larger eosinophilic tumor cells expressed smooth muscle actin (SMA), suggestive of focal myofibroblastic differentiation. Interestingly, tumors showed mixed positive staining for pan-CYTOKERATIN, which can be found in some soft tissue sarcomas such as undifferentiated pleomorphic sarcoma [45]. In addition, tumor cells showed negative staining for the vascular marker CD31 and for HMB45, thus excluding Kaposi sarcoma or other vascular tumors and clear cell sarcoma or neoplasms with perivascular epithelioid cell differentiation (PEComas), respectively. Collectively, the morphological and molecular features of these tumors are consistent with a diagnosis of undifferentiated pleomorphic sarcoma (UPS).

Since UPS tumors in humans are frequently associated with distant metastases, representing the leading cause of high morbidity and mortality of this tumor type [46,47,48,49], we used BLI to investigate whether our autochthonous models also exhibit metastatic dissemination. We injected SCID/Beige mice intramuscularly with MuLE lentiviruses expressing either shRNA-*Cdkn2a* + *Hras^G12V^*, shRNA-*Trp53* + *Hras^G12V^* or shRNA-*Trp53* + shRNA-*Pten* + *Hras^G12V^* and monitored tumor growth with BLI, sacrificing mice on an individual basis once the primary tumors reached approximately 10^9^ p/s in BLI. Prior to sacrifice and immediately following dissection, a final BLI was performed to allow identification of metastatic lesions in the liver and lung (Figure 1A–C), two common sites of UPS metastasis in humans [50,51]. The shRNA-*Trp53 + Hras^G12V^*-driven UPS model showed metastases in 100% of animals (*n* = 11) with a mean time to sacrifice of 72 days (Figure 1A). In contrast, shRNA-*Cdkn2a + Hras^G12V^*-driven tumors grew more rapidly and were sacrificed on average 40 days after injection (Figure 1B). These mice did not show metastases, possibly because the rapid growth of the primary tumor did not allow enough time for the spread and growth of tumor cells in distal organs. shRNA-*Trp53* + shRNA-*Pten* + *Hras^G12V^*-driven tumors developed with incomplete penetrance and grew more slowly, taking on average more than 100 days until sacrifice (Figure 1C). These mice surprisingly did not demonstrate BLI signals in distal organs.

To further characterize the invasive and metastatic phenotype of the shRNA-*Trp53 + Hras^G12V^*-derived UPS mouse model we generated cell lines from the primary tumor and by microdissection we generated cell lines from metastases in the lungs and/or the liver of 5 animals. All cell lines are hereafter identified with a unique number corresponding to the mouse they had been isolated from followed by the corresponding organ. For example, primary tumor cell line from mouse number 1 was named UPS 1 Pr, lung and liver cell lines from the same animal were labelled UPS 1 Lu and UPS 1 Li, respectively. We screened all cell lines by western blotting (Figure 1D) which revealed that 13 of 16 cell lines expressed high levels of H-RAS and lacked expression of the fibroblast marker DESMIN and muscle markers MYOGENIN and MYOD, consistent with these cell lines being tumor-derived. Three cell lines were excluded from further analyses as they exhibited a fibroblast-like phenotype in culture, expressed the fibroblast marker DESMIN and showed very low levels of expression of H-RAS, suggestive of fibroblast outgrowth of the culture rather than tumor cells.

We next asked whether cell lines isolated from metastatic sites exhibit cellular behavioral properties that were different to the properties of the cell lines isolated from the primary tumors. We therefore characterized three independent pairs of primary and liver metastasis cell lines with particular focus on their migratory and invasive properties by using the scratch and the trans-well migration assays. However, primary cell lines and their metastatic counterparts showed no significant differences in wound healing or invasion (Figure 1E,F and Appendix A). This result was further confirmed by in vivo allograft tumorigenicity assays (Figure 1G–I). Primary tumors and metastatic lesion-derived cell lines were transduced with a MuLE vector carrying eGFP and luciferase (pMULE SV40-eGFP-PGK-Luc) in order to achieve a comparable baseline luciferase expression level. After a period of time ranging from 9 to 20 days after subcutaneous injection of the primary and metastatic cell lines (Figure 1G), all mice showed metastatic dissemination to distal organs, namely liver, lung, brain and bone marrow (Figure 1H). Luciferase signal intensity from metastatic sites was obtained for every organ, which revealed that there were no obvious quantitative differences between the primary and metastatic cell line pairs in terms of metastatic propensity or homing to different organs (Figure 1I). The only difference was that luciferase intensity in the lungs of mice injected with the cell line UPS 7 Pr was significantly higher than the corresponding metastatic cell line UPS 7 Li. Our findings collectively show that the metastatic and aggressive phenotypes are already intrinsic in the primary tumor cell lines and are not an acquired feature of a very small subset of cells in the tumor that has been selected for in the metastatic lesions. Thus, our panel of cell lines represent a novel model of oncogenic *Hras^G12V^* driven metastasis which we employ for the therapeutic experiments described below.

### 3.2. Development of a Kras^G12D^-Driven Mouse Model of Metastatic Pancreatic Ductal Adenocarcinoma

The KPC (*K**ras^G12D^*-*Trp53^+/R172H^*
Cre-driven) mouse model of PDAC recapitulates many aspects of the human disease, including the histopathology and metastatic profile [35]. These mice harbor conditional *Kras^G12D^* and heterozygous *Trp53^R172H^* alleles that are expressed specifically in pancreatic cells under the control of the *Pdx1-Cre*-transgene. To generate primary tumor cell lines that represent an in vitro system of PDAC, primary cells were isolated from four different sites from a single 23 week old female KPC mouse (Figure 2A); two different regions of pancreatic tumor tissue differing in their stiffness (soft and solid) as well as from liver and spleen metastases. These cell lines were designated KPC Pr sof, KPC Pr sol, KPC Li and KPC Sp respectively. Cells in all cell lines grew predominantly as spindle-shaped mesenchymal-like cells while cells in the KPC Pr sof cell line additionally contained cells that grew as epithelial cell islands (Figure 2B). PCR analyses of genomic DNA isolated from the cell lines revealed the presence of the recombined *Kras^G12D^* allele in all four cell lines, however, while the KPC Pr sof cell line displayed an intact *Kras* wild type allele, the other three cell lines lacked this band, indicative of loss of the remaining wild type allele (Appendix A). Sequencing of the isolated PCR bands confirmed that the upper band represents the recombined allele and the lower band represents the wild type allele (Appendix A). It has been independently shown that loss of the *Kras* wild type allele is associated with metastasis in this model [52], suggesting that there is a selective advantage associated with this genetic alteration.

We next screened the cell lines for cancer stem cell markers by staining for CD44, CD24 and prominin-1 (CD133). In human pancreatic cancers, CD44^+^CD24^+^ESA^+^ cells have been found to be enriched for cancer stem cells (CSCs) verified by testing the tumor initiating capability of those cells in vivo [53]. CD133 has also been reported to be a marker for CSC in human pancreatic cancer [54]. The majority of cells in each of the four murine PDAC cell lines highly expressed CD24 and CD44 and we noted that some cells express CD44 at very high levels (Figure 2C). All cell lines exhibited a population of CD24^high^CD133^high^ cells that represented 8–9% of the entire population (Figure 2D). We conclude that our murine KPC cell cultures express markers that define human PDAC cancer stem cells.

To investigate whether growth factor signaling pathways are intact in these cell lines we performed western blotting on lysates from cells grown in the presence of serum with or without a 5 min stimulation with EGF. All key signaling molecules were detected in these lysates, including EGFR, BRAF, CRAF, AKT, MEK and ERK (Figure 2E,F), and EGF stimulation robustly increased the phosphorylation of MEK and ERK, and more weakly increased AKT phosphorylation (Figure 2F). These results demonstrate that growth factor mediated signal transduction functions as expected in the PDAC cell lines, suggesting that these cell lines represent a useful system for interrogating the effects of inhibition of downstream signaling in *Kras^G12D^* expressing cells.

### 3.3. Diverse Oncogenic RAS-Driven Mouse and Human Cancer Cell Lines Are Sensitive to Dual MEKi and ERKi Treatment

We next sought to take advantage of the clean genetics of our panels of cell lines from genetically engineered *Hras^G12V^*-driven UPS and *Kras^G12D^*-driven PDAC models to probe features of therapeutic sensitivity and resistance to inhibitors of MEK and ERK. We first selected three independent UPS cell lines with comparable growth rates, UPS 1 Pr, UPS 2 Pr and UPS7 Pr, and analyzed cell proliferation and viability when treated singly, doubly and triply with the following inhibitors: RO5126766 (RAF/MEK 1/2 inhibitor, hereafter abbreviated as RO), GDC-0623 (MEK1/2 inhibitor, hereafter abbreviated as GDC) and SCH772984 (ERK1/2 inhibitor, hereafter abbreviated as SCH). All single, double and triple combinations inhibited proliferation of all three cell lines measured after 72 h of exposure to the drugs (Figure 3A), while only the combinations RO + SCH, GDC + SCH and RO + GDC + SCH reduced viability across all cell lines (Figure 3B). Encouraged by these findings, we expanded the anti-proliferative assays to include additional *Hras^G12V^*-driven mouse UPS primary and metastatic cell lines (Figure 3C), *Kras^G12D^*-driven mouse PDAC primary and metastatic cell lines (Figure 3D), five established human cell lines from different tumor entities that harbor mutations in *KRAS, NRAS* or *BRAF* genes; the colorectal carcinoma cell lines HCT116 (*KRAS^G13D^*), RKO (*BRAF^V600E^*) and SW480 (*KRAS^G12V^*), the non-small cell lung carcinoma cell line NCI-H1299 (*NRAS^Q61K^*), the rhabdomyosarcoma cell line RD (*NRAS^Q61H^*) (Figure 3E), as well as two established human PDAC cancer cell lines (HPAF II and CAPAN-2 both harboring *KRAS^G12D^*) and two new primary patient-derived PDAC cell lines, namely B23 harboring *KRAS^G12D^* and B40 harboring *KRAS^G12V^*, that we generated in our institution (Figure 3F). In all cell lines, 72-h exposure to each single drug significantly reduced proliferation compared to DMSO (*p* < 0.0001). The magnitude of proliferative inhibition varied from approximately 43% to 97% for single treatments and between 60% and 108% for combined treatments depending on the drug and cell line. Overall, a common trend was observed among all cell lines that the combined agents had stronger anti-proliferative effects than the single agents, although the effect of combined therapy in relation to the corresponding single drug administrations reached statistical significance in only some of the combinations and cell lines (Figure 3C–F). Triple treatment was also performed for some cell lines but did not show any improved effect compared to the dual-drug combinations.

To further investigate the observation that dual MEKi and ERKi has stronger anti-proliferative effects than either drug alone, we performed a drug interaction assay. An 8 × 10 matrix combination dose–response screen of GDC and SCH was performed in 6 tumor cell lines (3 derived from our UPS mouse model and 3 from human cancers) to assess both single-agent activity and evaluate additive, synergistic, or antagonistic interactions across a range of doses. Absorbance values were used to determine the coefficient of drug interaction (CI) through the method of isoboles (Figure 3G,H). These analyses revealed that combined GDH+SCH treatment shows synergistic anti-proliferative activity (CI < 1) in all 6 cell lines tested. These results are encouraging in the context of future potential clinical translation as they demonstrate that the combined anti-proliferative effects of the drugs are likely to be present across ranges of concentrations of each individual drug, an important feature in the context of likely differences in the in vivo pharmacological properties of each agent.

### 3.4. Dual MEKi Plus ERKi Prevents the Emergence of Drug Resistance

Building upon these findings based on short-term drug testing, we next conducted long-term colony formation drug sensitivity assays, aiming to investigate possible resistance mechanisms. We focused on the most effective agents that emerged from the first set of experiments, namely the MEKi GDC and the ERKi SCH, and also included a second ERKi, namely LY3214996 (hereafter referred to as LY) which also inhibits both ERK1 and ERK2. Single treatments for 14 days with GDC, SCH and LY all reduced the number of colonies formed, with LY being less effective than SCH (Figure 4A). Double therapy with GDC+LY was more efficient at reducing colony formation than the single agents alone. However, while some colonies formed in most cell lines treated with single agents or with GDC+LY, demonstrating the presence or development of resistant cells, there were no colonies present in any of the cell lines treated with the GDC + SCH combination. To follow up on these observations that were based on clonogenic survival starting with relatively few cells, we next assessed survival of cells and the potential emergence of resistance in cultures beginning with larger numbers of cells. We performed long-term drug resistance assays by seeding 5 × 10^5^ cells in 6-cm plates and treating with 1 μM GDC, LY or SCH as single or double treatment combinations. Medium and inhibitors were refreshed every three days. Treatments were carried out for a maximum of four months and during this time any cells that reached confluence were split in a 1:5 ratio. In contrast to the strong inhibitory effects of the single MEKi and ERKi drugs on clonogenic survival when cells were plated at low density, these agents showed little effect when applied to more dense cultures of cells and all of the cells lines treated with individual drugs reached confluency within a few days to a week and continued proliferating after splitting, thereby generating resistant cell lines that were able to be continuously passaged in the presence of drug. At the time of splitting, we also generated replicate plates from each cell line and removed the drug from them to test whether they may be addicted to the presence of the drug, however we could not observe any drug addiction phenomenon since all cell lines continued proliferating in the absence of the drug (data not shown). In contrast to the single treatments, many cell lines treated with combinations of drugs never reached the splitting stage and cells underwent growth arrest or cell death. Crystal violet staining of cells seeded in parallel after 14 days of treatment provides a visualization of the differing cell responses to long-term treatment (Appendix A). Experiments were stopped at the stage when we could not observe any remaining living cells in the cell culture plates and these cells were classified as never giving rise to resistance. These experiments are summarized in Figure 4B as a color code in which red shades represent the time taken to reach resistance and blue represents conditions in which resistance did not arise. Dual treatment with the two ERKi LY+SCH prevented resistance in 6 of 16 cell lines and dual treatment with the MEKi/ERKi combination GDC+LY prevented resistance in 15 of 16 cell lines. The MEKi/ERKi dual combination of GDC + SCH was the most effective and prevented the emergence of resistance in all of the cell lines.

Collectively, our results using diverse mouse and human oncogenic RAS-pathway driven cancer cell lines provide evidence that the strategy of vertical pathway inhibition using combined MEKi + ERKi synergistically inhibits cellular proliferation and prevents the emergence of drug resistance.

### 3.5. Testing Dual MEKi Plus ERKi Therapy In Vivo

We next sought to test the two most promising drugs, GDC and SCH, for their activities in vivo in the shRNA-*Trp53 + Hras^G12V^*-driven UPS model. First experiments involving the commercially available formulation of SCH772984 from Selleckchem (S7101), following the exact protocols used for solubilizing and injecting this compound that were described in the original publication describing the invention of this molecule [21] revealed problems with solubility, necessitating the injection of very large volumes to achieve the published doses. These volumes were not tolerated by mice. Upon enquiry, we were subsequently informed by the authors of this publication that they in fact had used an HCl salt formulation of the molecule, SCH772984-HCl. We therefore had this molecule synthesized by Selleckchem, which solved the solubility issues and allowed a limited number of pilot experiments to be conducted.

We injected 3-week-old SCID mice with MuLE lentiviruses expressing shRNA-*Trp53 + Hras^G12V^* + luciferase and waited until the tumors reached bioluminescence levels between 5 × 10^8^ and 1 × 10^9^ p/s (approximately within 50 days after injection) in order for the therapeutic treatment to be initiated. Mice were treated every day for 4 days with either vehicle (*n* = 2 mice), GDC0623 (*n* = 2 mice, 40 mg/kg daily by oral gavage), SCH772984-HCl (*n* = 3 mice, 50 mg/kg twice daily by intraperitoneal injection) or with both GDC0623 + SCH772984-HCl (*n* = 2 mice, using the schedules for the individual therapies). These doses had previously been described as maximum tolerated doses in single treatment experiments [21,55]. Immunohistochemical stainings (Figure 5A) revealed that the GDC and GDC + SCH treatments reduced levels of the MEK target phospho-T202/Y204-ERK, demonstrating on-target activity of GDC in vivo. GDC and SCH single treatments each partly reduced levels of the ERK target phospho-S265-FRA1 and these levels were further reduced in tumors treated with GDC + SCH, verifying the on-target activity of SCH and demonstrating a stronger inhibition of RAS signaling by the dual therapy.

Based on these findings, we sought to test whether GDC + SCH treatment could inhibit tumor growth in longer term therapy experiments. We intramuscularly injected 3-week-old SCID mice with MuLE lentiviruses expressing shRNA-*Trp53 + Hras^G12V^* + luciferase and waited until the tumors reached bioluminescence levels between 5x10^7^ and 1 × 10^8^ p/s (within 12–50 days after injection) before initiating therapy involving a vehicle control group (*n* = 5 mice) and a group (*n* = 7) treated with GDC0623 (40 mg/kg daily by oral gavage) + SCH772984-HCl (50 mg/kg twice daily by intraperitoneal injection). Mice were treated for a maximum period of 6 weeks according to a 5 days-on/2 days-off schedule. Termination criteria were bioluminescence signal intensity higher than 5 × 10^9^ p/s or deteriorating health of the animal. All five of the control-treated mice developed tumors, 2 of which were sacrificed at day 12 and 30 after start of treatment because the tumors quickly reached the bioluminescence end-point, while the remaining 3 mice were sacrificed with tumors at the end of the 6 weeks of treatment. Longer term GDC + SCH therapy was not well tolerated by the animals. While the mice did not lose weight (Figure 5B), and despite twice daily monitoring of their health status, sudden death or rapid development of a moribund state required urgent termination due to suffering. These deaths and acute sickness appeared without any apparent warning signs. Six of the seven mice died or were sacrificed 12, 15, 16, 26, 32 or 33 days after start of treatment and the remaining mouse was sacrificed in a healthy state at 38 days of therapy. Autopsy did not reveal any obvious pathologies that would account for the deaths or moribund state. Interestingly, tumor growth was observed in only one mouse throughout the treatment while three mice exhibited stable BLI signals and three showed a decrease in the BLI signals. In all but one of the seven treated animals, we were unable to identify a macroscopically visible tumor (Figure 5C). Detailed histological analyses of dissected muscles however revealed the presence of small nests of cells with features of cancer cells in 4 of 7 treated mice (Appendix A). Immunohistochemical staining of the isolated tumor from the GDC + SCH group could not be performed because the mouse was found dead and we were unable to dissect it anymore. Tumor nests in dissected muscles of treated mice were however negative for phospho-T202/Y204-ERK, yet positive for phospho-S265-FRA1 (Figure 5D), suggestive of reactivation of ERK activity or activation of other kinases involved in FRA1 regulation [56]. The absence of phospho-S265-FRA1 in short term therapy experiments but present in nest of tumor cells in long term therapy experiments is consistent with the potential development of drug resistance.

With the obvious caveat of the long term toxicity of this particular therapeutic regime, these observations are consistent with an anti-tumor activity of the dual MEKi + ERKi treatment. Future experiments would need to systematically assess tolerable doses, delivery schedules and combinations of different MEKi and ERKi compounds to identify to best strategy to follow in further pre-clinical and clinical development.

### 3.6. Acquired Resistance to Single MEKi or ERKi Overcomes Susceptibility to Dual Inhibition

Given our observations of resistance to single agent therapy in cell culture, and hints from the in vivo studies that resistance might also be an issue in the setting of combined therapy, we next sought to use our large series of mouse and human cancer cell models to gain further insight into the molecular features of resistance. We first investigated whether the pre-existing development of resistance to one RAS-pathway targeting agent may affect sensitivity to another RAS-pathway targeting agent and in particular whether this might abolish the ability of combination treatment to block the emergence of drug resistance. In order to see whether cell lines resistant to MEKi still retain sensitivity to ERKi alone or in combination with MEKi and *vice versa*, we treated established resistant cells for the MEKi GDC (denoted as GDC-res) with the ERKi SCH or with GDC + SCH and we treated resistant cells for the ERKi SCH (denoted as SCH-res) with the MEKi GDC or with GDC + SCH for up to 4 months until cells gained resistance or we could not observe any remaining living cells in the plates (Figure 6A). Figure 6B summarizes these analyses and shows that cells that are resistant to either one or the other single agent were not affected by the presence of the other agent since they all rapidly developed resistance (range between 7 and 30 days after start of treatment). In contrast to the findings made in the parental cell lines, dual inhibition through combined administration of GDC + SCH to cell lines resistant to single GDC or SCH showed a much lower overall efficacy of the treatment in terms of preventing resistance (Figure 4B). Notably, only four cell lines (UPS 1 Pr GDC-res, UPS 7 Li GDC-res, UPS 7 Li SCH-res and RKO SCH-res) were very sensitive to the combined therapy and died within 20 days of treatment, whereas 26 of the cell lines started growing out with kinetics and morphology indistinguishable from the starting cell population after a relative short time of adaptation. Interestingly, among the cell lines that initially grew well in the presence of the dual inhibition, 20 of them went through a phase during which they had stopped proliferating and reduced their metabolism to very low levels, judging from the color of culturing medium. In order to assess their phenotype, we trypsinized and re-plated them at day 55 after treatment and observed two different reactions, one being cell death in 3 out of 20 cell lines and the other being recovery of proliferation fitness leading to the development of resistant cell lines in 17 out of 20 cases. Overall, our findings show that, conversely to what other studies have shown for *KRAS*-driven cell lines [31], ERK inhibition cannot overcome acquired resistance to MEK inhibitors as single agent nor as combined therapy. Importantly, extrapolating to the clinic, our findings strongly suggest that in order to minimize the chances of the emergence of drug resistance a dual MEKi and ERKi inhibition approach should be applied as first line therapy rather than applying a single MEKi or ERKi therapy.

### 3.7. Kinome Profiling Shows Large Variation and Complex Signalling Rewiring in Resistant Cell Lines

To gain a first insight into potential molecular mechanisms underlying resistance, we studied the effects of different MEKi and ERKi on the *Hras^G12V^*-driven UPS 1 Pr cell line. We treated parental UPS 1 Pr cells for 24 h with either RO, GDC, LY, SCH or RO + GDC and in parallel treated UPS 1 Pr cell lines that had previously developed resistance to these agents. (Appendix A). In parental cells, as expected, the two MEKi RO and GDC reduced the phosphorylation of the MEK target phospho-T202/Y204-ERK and of the downstream ERK target phospho-S265-FRA1 while the two ERKi inhibitors LY and SCH did not affect phospho-T202/Y204-ERK but blocked phosphorylation of phospho-S265-FRA1. Interestingly however, RO-resistant cells, GDC-resistant cells, and RO + GDC-resistant cells all failed to downregulate phospho-T202/Y204-ERK in response to the relevant drug, while RO-resistant and RO + GDC-resistant cells failed to downregulate phospho-S265-FRA1, yet this response was intact in GDC-resistant cells. Similarly, SCH-resistant cells normally downregulated phospho-S265-FRA1 while LY-resistant cells did not. Thus, even within one cell line and at the level of analysis of just two kinases, there are differences in the patterns of sensitivity that have arisen between different inhibitors of the same targets, suggesting that resistance is likely to represent a highly complex process.

We therefore adopted a more systematic approach to profile molecular changes at the level of kinome activity. Previous studies in oncogenic *KRAS*- or *Kras*-driven human or mouse PDAC cell lines revealed that genetic loss of KRAS function leads to rewiring of cellular signaling networks that are associated with continued cellular proliferation in the absence of the driver oncogene [57,58]. To investigate whether a similar phenomenon might underlie resistance to downstream pharmacological inhibition of oncogenic RAS-pathway signaling we employed a multiplex kinase assay to ask whether MEKi and ERKi resistant cell lines are characterized by distinct kinome signatures with the idea of potentially being able to identify resistance-specific kinase signaling alterations that could be therapeutically actionable. To this aim we selected nine cell lines; six of mouse origin (three from primary tumors and three from metastatic lesions) representing both oncogenic *Hras*- and *Kras*-driven tumors as well as three of human origin harboring either *KRAS*, *NRAS* or *BRAF* oncogenic mutations, that were resistant to both single MEKi and ERKi (GDC-res and SCH-res respectively) and that had developed secondary resistance to the GDC + SCH combination in a comparable time-span (20–35 days) (generated according to the workflow depicted in Figure 6A). Protein lysates from resistant cell lines grown in the presence of the drug(s) to which they had developed resistance and their corresponding untreated parental cell lines were profiled for phosphotyrosine kinase (PTK) and serine-threonine kinase (STK) activities on PamChip^®^ peptide microarrays. These 3D microarrays are spotted with peptides (phosphosites) that represent kinase targets and the extent of their phosphorylation in an in vitro assay allows inference of the activity of upstream kinases in the lysate. One PamChip^®^ per cell line was employed in order to allow direct comparison of parental untreated, GDC-resistant, SCH-resistant and GDC + SCH-resistant cells. Data analysis was performed with the BioNavigator63 software, which allowed upstream kinase analysis (UKA), thereby providing information on which kinases are responsible for the differences in peptide phosphorylation between the cell lines. Full data is provided in Appendix A. Changes in kinase expression were calculated as mean kinase scores and quantified as log_2_Fold Change (LFC). Negative and positive values denote downregulation and upregulation of each kinase in resistant cells compared to their parental counterpart, respectively (Figure 7A,B and Appendix A). Somewhat unexpectedly, given that many of these cell lines are derived from genetically-defined mouse models with “simple” tumor genetics, we observed large variations in the overall patterns of STK and PTK kinase activities when comparing the parental and resistant cells within an individual cell line as well as across all related or unrelated cell lines. In some resistant cell lines global kinase activities were higher than in parental cells, in some they were lower and in others there were both up and down regulated kinases. In general, there were more alterations in STKs than PTKs when comparing resistant cells to parental cells (Figure 7A,B and Appendix A). Heatmap analyses at the level of individual kinases based on absolute changes in kinase activity (Figure 7C and Appendix A) or based on z-scores of relative kinase activity to attempt to correct for global changes in kinase activities within each particular sample (Figure 7D and Appendix A), failed to reveal kinases whose activities were commonly altered across all samples. These results point towards a cell line- and drug-dependent signaling rewiring and highlight that there is a large degree of complexity and variability behind development of drug resistance.

Despite the lack of consistent kinase activity signatures across all cell lines we reasoned that altered kinase activities in individual cell lines, or subsets of cell lines, may contribute to the resistance phenotype and may therefore represent targets for therapy in resistant cells. Overlapping upregulated kinases with LFC mean kinase score greater than or equal to 1.5 were identified in only 5 of the 27 resistant cell lines (Figure 8A and Appendix A). Eleven cell lines exhibited overlaps in downregulated kinases (LFC < 1.5) (Appendix A and Appendix A). We next analyzed the lists of upregulated kinases for which specific inhibitors are available and identified PKAα, MAP4K4 (HGZ/ZC1) and PKG2 among the upregulated STKs as putative therapeutic targets (Figure 8B–D). We then reanalyzed all cell lines that had been employed for the kinome activity assays regardless of the LFC mean kinase scores associated to PKA, MAP4K4 and PKG2 and performed a long-term drug sensitivity assay. Fifty thousand cells per well were seeded in 12-well plates and cells were treated with one of two inhibitors targeting PKA (H 89 2HCl, hereafter named P1 and PKA inhibitor fragment (6–22) amide, hereafter named P2), one of two inhibitors targeting MAP4K4 (PF-6260933, hereafter named M1 and GNE-495, hereafter named M2) or one inhibitor targeting PKG2 (AP C5, hereafter named PK). These new inhibitors were administered to cells either individually or in combination with either GDC, SCH or GDC + SCH in the relevant resistant cell lines in order to test whether the presence or absence of the MEKi and/or ERKi affected sensitivity to the new kinase inhibitors. Medium and inhibitors were refreshed every three days and treatments were carried out for a maximum 38 days. During this time all cells had reached confluence and continued growing after splitting in a 1:5 ratio, thereby giving rise to resistant cell lines (Figure 8E). In general, all parental cells continued to grow in the presence of the newly tested inhibitors, while anti-proliferative effects that lead to delayed resistance for approximately 1 month could be observed in particular in some of the SCH-resistant and GDC + SCH-resistant cell lines. For example, GDC + SCH-resistant cells were inhibited to different degrees by all of the PKA, MAP4K4 and PKG2 inhibitors and these effects were surprisingly diminished by co-treatment with GDC and SCH. GDC + SCH resistant UPS 2 Pr and UPS 9 Li cells were inhibited by the MAP4K4 inhibitors and these effects were abolished by co-treatment with GDC and SCH. These results suggest that the evolution of resistance in these three particular cell lines involved a signaling rewiring that led to a partial proliferative dependency on MAP4K4 kinase activity but that the presence of the GDC and SCH drugs overrides this dependency. Interestingly and in contrast, the SCH-resistant UPS 2 Pr cell line is not inhibited by MAP4K4 inhibition alone but is inhibited when SCH is co-administered. Thus, the same parental cell line developed different patterns of proliferative dependency on MAP4K4 during the course of development of resistance to MEKi or to MEKi plus ERKi. Surprisingly, we were unable to identify any relationships between upstream kinase activity scores and patterns of sensitivity to the different inhibitors. For example, the GDH+SCH resistant cell lines that were sensitive to MAP4K4 inhibition did not exhibit higher MAP4K4 expression than the parental cells. This discovery was therefore made serendipitously. These results nonetheless provide proof-of-principle that despite the variability in overall patterns of kinome signaling it is possible to identify new pharmacological dependencies in drug resistant cells.

## 4. Discussion

Oncogenic mutations in *RAS* family genes or in *BRAF* arise frequently in numerous types of metastatic human tumors for which, in the advanced stage, curative therapeutic options are not currently available. To address this clinical problem, in this study we used reverse genetic tumor engineering in mice to develop new models of oncogenic *Hras^G12V^*-driven UPS metastasis and of *Kras^G12D^*-driven PDAC metastasis. We took advantage of the clean genetics of these models to uncover insights into patterns of therapeutic sensitivity and resistance to targeted inhibition of downstream MEK and ERK signaling in RAS-driven tumors. Through analyses of a variety of human *KRAS, NRAS* or *BRAF* mutant cell lines from different tumor entities (CRC, NSCLC, PDAC, RMS) we further showed that these principles also apply to the complex genetic backgrounds that arise during human RAS-pathway driven cancer development.

Molecularly targeted therapeutic approaches for oncogenic RAS-RAF-MEK-ERK signaling initially targeted RAF and MEK, either as single or combined agents. The observation that ERK reactivation is often a driver of drug resistance upon RAFi and/or MEKi led to the development of inhibitors of ERK. In this study we compared the effects of two MEKi (GDC-0623 and RO5126766) and two ERKi (SCH772984 and LY3214996) as single and combined agents. While these drugs nominally have similar features in terms of mechanism of action and/or inhibition strength, as well as selectivity for their targets, different inhibitory efficiencies were observed in our in vitro experiments. The MEKi GDC-0623 outperformed MEKi RO5126766, while the ERKi SCH772984 greatly outperformed ERKi LY3214996 in terms of effects on short-term proliferative inhibition, as well as on emergence of drug resistance. Since we observed that cultures of cells treated with single MEKi or ERKi rapidly developed resistance and continued proliferating, we conducted longer term assays that identified that combined treatment with MEKi GDC-0623 and ERKi SCH772984 was the only combination that did not allow the emergence of drug resistance in all of the tested mouse and human cell lines. These results are consistent with a previous study that showed that combined MEK and ERK inhibition had better therapeutic effects and more deeply and durably suppressed pathway signaling than single MEK or ERK inhibition [33]. Importantly, we demonstrated that for the majority of cell lines, the prior development of resistance to a single MEK or ERK inhibitor allowed the development of resistance to the combination treatment.

Extrapolating to the clinical setting, these findings imply that combined targeted inhibition of MEK and ERK could be used as first-line therapy to achieve better inhibitory effects on the oncogenic signaling pathway, more efficient anti-proliferative responses and importantly to pre-empt the development of resistance to single agents. Indeed, a phase I clinical trial involving ERKi MK-8353 in combination with MEKi Selumetinib (NCT03745989) in patients with histologically or cytologically confirmed diagnosis of advanced solid tumor is currently underway. In this context, we provide proof-of-principle evidence that combined GDC-0623 and SCH772984 therapy is efficacious by showing that short term combination treatment efficiently inhibits MEK and ERK activities in the autochthonous *Hras^G12V^*-driven UPS model and that long term therapy inhibits the evolution of macroscopic UPS tumors. We however note that this therapeutic regime in mice also caused as-yet-undefined toxicities. Future studies should focus on optimizing combinations of doses and timing of drug delivery. During the course of this study, the “tool compound” SCH772984 has evolved into the clinical derivative compound MK-8353 that has better pharmacological properties and a good tolerability profile in phase I clinical studies [20]. Future mouse therapy experiments should incorporate this molecule as the lead ERKi candidate. A recent study using mouse models also discovered that triple or quadruple pharmacological vertical inhibition using low doses of combinations of EGFR, RAF, MEK and ERK achieves efficient pathway inhibition [50,59], suggesting that further dose optimization may potentially be undertaken to maintain on-target effects and eliminate toxic effects of combinatorial MEK and ERK inhibitory therapy. It has also been demonstrated that intermittent therapy schedules involving triple RAFi, MEKi and ERKi are more effective against *BRAF^V600E^*-driven tumors and are less toxic than continuous schedules in mouse models [32], suggesting that intermittent therapy regimes could also be employed to minimize potential toxicities. However, a first clinical test of this concept demonstrated that continuous dosing was superior to intermittent dosing of combination BRAF/MEK inhibitor therapy for advanced melanoma [60], questioning whether this concept from mouse models may translate to the human disease setting.

Increasing evidence suggests that mechanisms of resistance to downstream pathway inhibition differ between *BRAF* mutant tumors and *RAS* mutant tumors. Resistance to RAFi and/or MEKi in oncogenic *BRAF*-driven cells is a slower process that frequently involves the selection of small numbers of resistant cells in a population that often harbor a specific genetic alteration in one of a variety of genes that abrogates the sensitivity to the inhibitor. For example, in a *BRAF^V600E^*-mutant melanoma model the development of resistance to sequential treatment with the RAFi Vemurafenib, the MEKi Trametinib and the ERKi SCH772984 was shown to be due to progressive increase in *BRAF* copy number [32]. Other resistance mechanisms include the RAFi induced alternative splicing of *BRAF^V600E^* leading to the production an N-terminally truncated and strongly dimerizing oncoprotein, which is no longer inhibited due to the phenomenon of negative allostery, or the rapid reactivation of various RTKs that are silenced by ERK mediated negative feedback loops [61]. In contrast, we observe that resistance to single MEKi or ERKi treatment of oncogenic RAS-driven mouse and human cancer cells frequently occurs within days to a week. The timing of the emergence of this resistance is not compatible with the concept that resistance is driven by the outgrowth of a small sub-population of cells. Resistance appeared to generally involve a large fraction of the cell population which resumed proliferation, albeit often at a slower rate, sometimes with altered morphology and with apparently altered metabolism as indicated by the color of the medium. Similar conclusions were drawn from genetic studies involving the knockout of oncogenic *KRAS* or knockdown of *Kras* to genetically model the effects of a theoretical ideal KRAS inhibitor in human and murine PDAC cells [57,58]. These studies showed that many PDAC cell lines are not addicted to oncogenic KRAS and that they can rapidly adapt cellular signaling networks as non-genetic, cell population-level mechanisms of resistance to inhibition of KRAS function. Our in vivo therapeutic studies provide hints that resistance may also emerge in the context of dual MEKi and ERKi therapy. While short term dual GDC + SCH therapy effectively reduced phospho-ERK and phospho-FRA1 in UPS tumors, indicative of suppression of the kinase activities of MEK and of ERK, respectively, in the longer term therapeutic studies we identified microscopic nests of tumor cells that displayed no phospho-ERK but abundant phospho-FRA1, suggestive of reactivation or bypass of ERK activity. This may theoretically be due to non-optimal pharmacology where temporal variations in the intra-tumoral concentration of the ERK inhibitor might not sufficiently inhibit ERK, ultimately leading to development of resistance. While it will be important to determine whether improved pharmacological scheduling with MK-8353 might address this issue, we believe that it is also likely that similar issues of temporal variations of incomplete pharmacological inhibition are likely to be present in all pharmacological schedules involving MEK and ERK inhibitors and that drug resistance is likely to remain a clinical issue. Indeed, in another study involving *Kras^G12D^*-driven genetically engineered mouse models of NSCLC and PDAC, dual MEK (Cobimetinib) and ERK (GDC-0994) inhibition reduced tumor growth and increased progression-free survival, but the animals still died of the tumors, indicative of the emergence of drug resistance [33].

We therefore sought to better molecularly characterize the mechanisms of resistance to single and combined MEKi and ERKi in oncogenic KRAS-, HRAS- or BRAF-driven cancer cells by focusing analyses on potential rewiring of intracellular signaling cascades. Inspired by a recent study showing that screening the activities of more than 60 kinases in cell lines and in human tumor samples allowed the identification of kinases that are therapeutically relevant in the context of *BRAF^V600E^*-driven tumors [62], we used a phosphosite-specific kinase activity array system to assay the activity of tyrosine kinases towards 196 target peptides and of serine/threonine kinases towards 144 target peptides in nine different sets of mouse and human parental, MEKi-, ERKi- or MEKi/ERKi-resistant cells. These kinome-wide activity assays revealed that the patterns of rewiring of signaling activity in the various resistant cells are highly heterogeneous with limited overlap within and between cell lines. While follow up assays to assess whether any of the upregulated kinase activities might represent novel therapeutic targets in resistant cell lines were limited by the availability of specific inhibitors to the identified kinases, we identified that proliferation and development of drug resistance in three of the nine tested MEKi/ERKi-resistant cell lines were inhibited by two independent inhibitors of MAP4K4 but these drugs did not inhibit the parental or single MEKi- or ERKi-resistant cell lines. This finding provides proof-of-principle that it is possible to uncover novel pharmacological sensitivities that are specific to MEKi/ERKi-resistant cells. Interestingly, MAP4K4 (also known as HGK or NIK) was previously reported to be responsible for ERK1/2 activation in lung adenocarcinomas via inhibition of protein phosphatase 2 activity [63], providing a possible molecular explanation for the dependence of some MEKi/ERKi-resistant cell lines on MAP4K4 activity. The panels of parental and resistant mouse and human oncogenic RAS-driven cell lines that we have established in this study will represent powerful experimental systems enabling larger-scale chemical and genetic screening experiments to search for additional proteins or signaling pathways that may be able to be targeted therapeutically to prevent or revert resistance to single or combined MEKi and ERKi in oncogenic RAS-driven cancers.

## 5. Conclusions

We conclude that the development of resistance of oncogenic RAS-driven cancer cells to single MEK or ERK inhibitors can be prevented by up-front, simultaneous treatment with both drugs together. However, if single-agent resistance arises, subsequent dual MEK and ERK inhibitor treatment in many cases is unable to inhibit proliferation. Drug resistant cells exhibit heterogeneous patterns of re-wiring of kinase signaling and these kinases may represent new therapeutic targets in resistant cells.

## Figures and Tables

**Figure 1 cancers-13-01852-f001:**
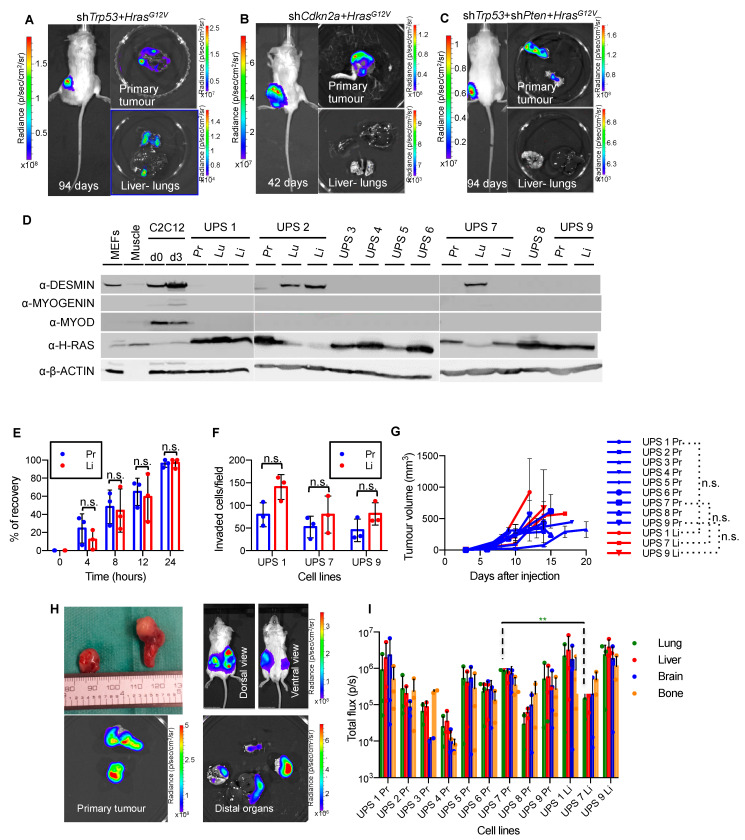
Establishing an *Hras^G12V^*-driven autochthonous metastatic UPS model. (**A**–**C**)**.** Representative bioluminescence imaging of living mice and of primary tumors, lungs and livers of tumor-bearing SCID/Beige mice induced by intramuscular injection of MuLE viruses expressing luciferase plus either shRNA-*Trp53* + *Hras^G12V^* (**A**) shRNA-*Cdkn2a* + *Hras^G12V^* (**B**) or shRNA-*Trp53* + shRNA-*Pten* + *Hras^G12V^* (**C**). (**D**). Protein immunoblotting for the indicated antibodies on cells isolated from primary UPS tumors (Pr) and metastatic lesions from lungs (Lu) and livers (Li), as well as on mouse embryo fibroblasts (MEFs), mouse muscle and C2C12 myoblast cells at day 0 and day 3 of myogenic differentiation as controls. VINCULIN is used as loading control. (**E**). Time course of recovery of wounds induced by scratch assay on confluent cells. Experiments were performed on UPS 7 Pr and UPS 7 Li. Results are representative of similar experiments performed using UPS 1 Pr and UPS 1 Li and using UPS 9 Pr and UPS 9 Li (Appendix A). Mean ± std. dev. are shown (*n* = 3). In all matching pairs of cells, the differences between primary tumor and metastatic cell lines were not statistically significant (*p* > 0.05, unpaired t-test Holm-Sidak method). (**F**). Overnight trans-well migration assay. Data is derived from analyses of ten independent fields of 1.5 mm^2^ per membrane and is depicted as mean ± std. dev. of 3 biological replicates. Differences were not statistically significant (*p* > 0.05, two-way ANOVA, Sidak method). (**G**). Growth curves of primary tumors arising from subcutaneous injection of the indicated primary and metastatic cell lines. Differences between growth rates of tumors arising from metastatic lesions over their corresponding primary tumor cell lines were not statistically significant (n.s., Welch’s *t*-test). (**H**). Example of primary tumor and metastatic lesions arising within 20 days after subcutaneous injection of cells isolated from *Trp53 + Hras^G12V^* primary tumor. Distal organs include lungs, liver, brain and femur (bone marrow). (**I**). Luciferase intensity from metastatic lesions arising from subcutaneous allografts of primary tumor or metastatic lesion cells. Mean ± std. dev. of each organ from 3 injected mice per cell line are shown. Luciferase intensity in lung arising from cell line UPS 7 Pr was significantly higher than the luciferase intensity in the lung from its metastatic counterpart (UPS 7 Li) (** *p* < 0.01, unpaired *t*-test Holm-Sidak method). All other comparisons were not statistically significant.

**Figure 2 cancers-13-01852-f002:**
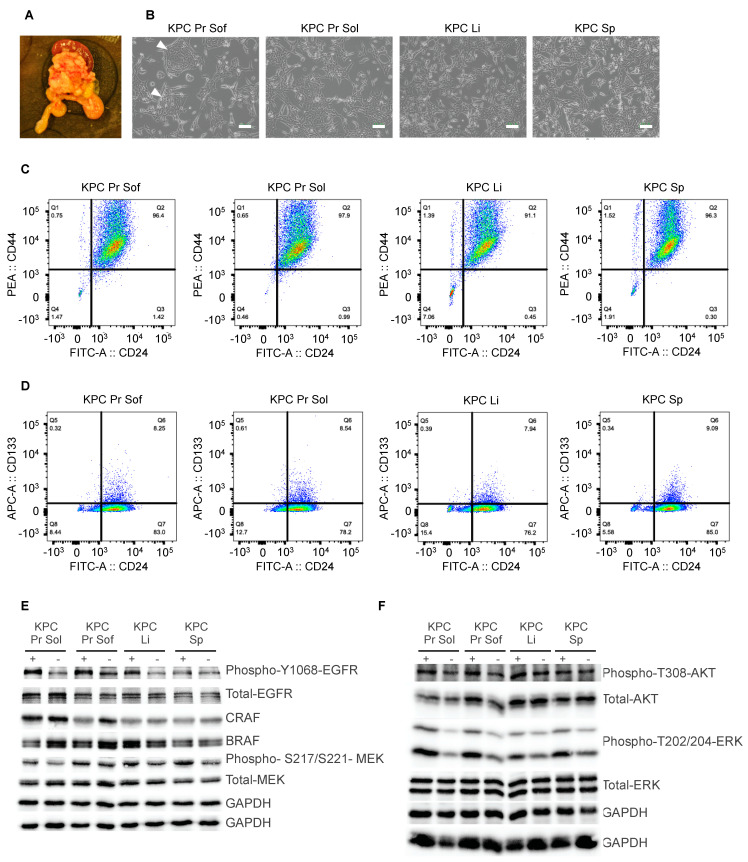
Establishing a *Kras^G12D^*-driven mouse model of metastatic PDAC. (**A**). Picture of primary tumor and spleen from which the cell lines were generated. (**B**). Representative images of primary cells four passages after isolation. White arrows indicate epithelial islands which were detectable only in the KPC Pr Sof cell line. Scale bars = 100 μm. (**C**,**D**). Flow cytometry analysis of CD44, CD24 and CD133 expression in pancreatic primary tumor and metastatic cell lines. 50,000 cells per cell line were analyzed. CD44 versus CD24 expression is shown after gates were set on the negative control (**C**) or on the single-stained CD24 control (**D**). (**E**,**F**). Protein immunoblotting for the indicated antibodies on cells isolated from primary KPC tumors (Pr Sof and Pr Sol) and metastatic lesions from liver (Li) and spleen (Sp) with or without a 5-min stimulation with EGF (10 ng/μL)prior to protein isolation. GAPDH blotting represented a loading control on the replicate blots used in this figure. Samples were run on replicate blots that were probed with different antibodies. Two different GAPDH loading control blots were therefore conducted.

**Figure 3 cancers-13-01852-f003:**
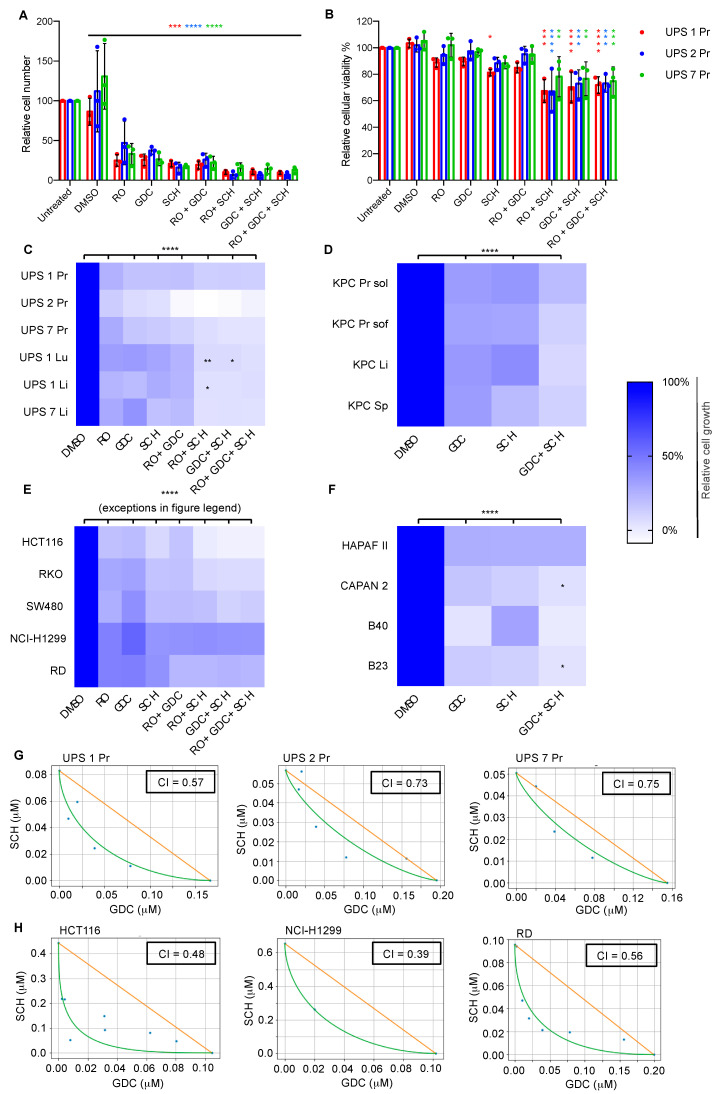
Oncogenic RAS-driven mouse and human cancer cell lines are sensitive to dual MEKi and ERKi treatment. (**A**,**B**). Relative cell number (**A**) and cellular viability (**B**) assessed on day three of treatment of the indicated UPS cell lines with the indicated inhibitors (1 μM). Results are normalized to the untreated condition, mean ± std. dev. of 3 biological replicates is shown and statistical differences compared to treatment with DMSO are shown. Two-Way ANOVA, Turkey’s method: * 0.05 < *p* < 0.0332, ** 0.0332 < *p*< 0.0021, *** 0.0021 < *p* < 0002, **** *p* < 0.0001. (**C**–**F**). Short-term (3 days) response of UPS (**C**), KPC (**D**) and human CRC, NSCLC, RMS (**E**) and PDAC (**F**) cell lines to the indicated inhibitors (1 μM). Mean of three independent experiments is shown, each of which involved three technical replicates. Differences of treatments compared to DMSO were all statistically significant, Dunnett’s multiple comparison test (2way ANOVA): * 0.05 < *p* < 0.0332, ** 0.0332 < *p* < 0.0021, *** 0.0021 < *p* < 0002, **** *p* < 0.0001. An exception in panel E to the high significance is cell line NCI-H1299 where cell growth after treatment with GDC was reduced to a less significant extent (**). Statistical significance of the double treatment is shown only when significant over both the respective single treatments. Triple treatment was more effective than double treatment only when compared to RO + GDC, which was ineffective on its own (**C**). Squares without significance stars were all non-significant. No statistically relevant differences were measured in the efficiency of treatments between cell lines from primary tumors and corresponding metastatic lesions (**C**). (**G**,**H**). Curves of interaction between SCH and GDC with respective synergistic score (CI) in the indicated mouse (**G**) and human cell lines (**H**). X and Y axes represent both the fixed concentration of the individual drug as well as the 72 h anti-proliferative IC50 value of that drug at the fixed concentration of the other drug.

**Figure 4 cancers-13-01852-f004:**
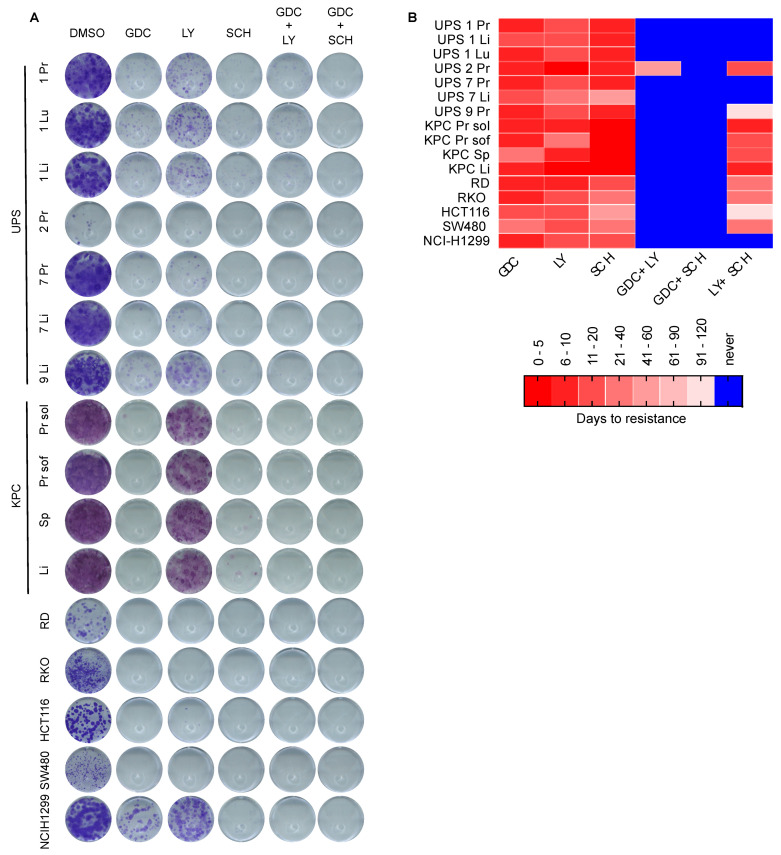
Dual MEKi plus ERKi prevents the emergence of drug resistance. (**A**). Examples of colony forming assays of the indicated cell lines cultivated for 14 days with the indicated inhibitors (1 μM). (**B**). Long-term drug resistance assay. The indicated sub-confluent cell lines (5 × 10^5^ cells/6 cm plate) were cultivated with the indicated inhibitors (1 μM). Days to resistance represents the timepoint at which cells were first split and continued proliferation thereafter, cells that did not survive the treatments or proliferate within 4 months are depicted in blue.

**Figure 5 cancers-13-01852-f005:**
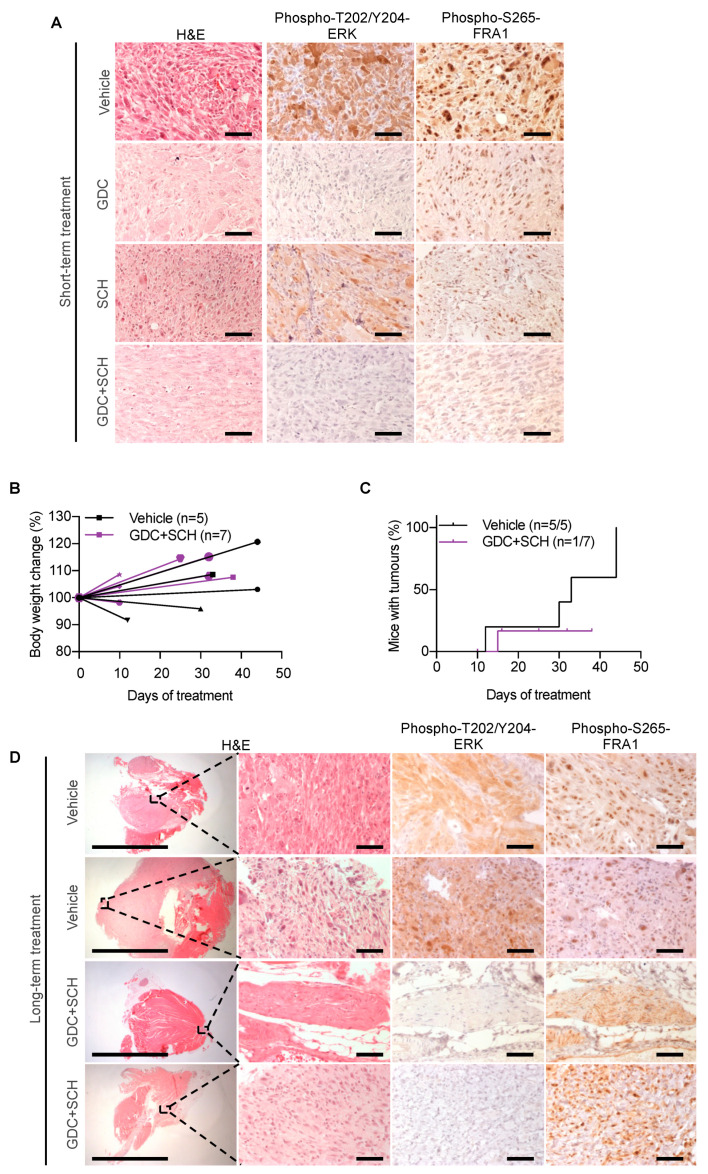
Testing dual MEKi plus ERKi therapy in vivo. (**A**) H&E and immunohistochemical stainings on tumor samples after short-term treatment with vehicle, GDC, SCH or GDC + SCH. Stainings are representative of 2, 2, 3 and 2 independent tumors of each treatment respectively. (**B**) Body weights trend/distribution between start and end of long-term therapy. Each line represents one mouse. 5 mice were treated with vehicle and 7 were treated with GDC + SCH. (**C**) Kaplan Meier curves used to represent percentage of mice with tumors throughout the course of long term treatment. 5 out 5 mice treated with vehicle developed tumors, whereas only one mouse out of 7 treated with GDC + SCH developed tumor. (**D**) H&E and immunohistochemical stainings on tumor samples after long-term treatment with vehicle or GDC + SCH. Two independent examples per treatment are shown. Left-end panels show representative low-magnification pictures of tumors developing in mice treated with vehicle and tumor nests in mice treated with GDC + SCH. Scale bar = 5 mm for the low-magnification pictures, while scale bar = 50 μm for the high-magnification pictures. Higher magnification IHC stainings in A and D were performed using the indicated antibodies against the effector signaling marker of the RAS pathway phospho-T202/Y204-ERK and one of its downstream targets phospho-S265-FRA1. Vehicle-treated samples were used as positive controls for both phospho-T202/Y204-ERK and phospho-S265-FRA1 given their RAS-driven origin.

**Figure 6 cancers-13-01852-f006:**
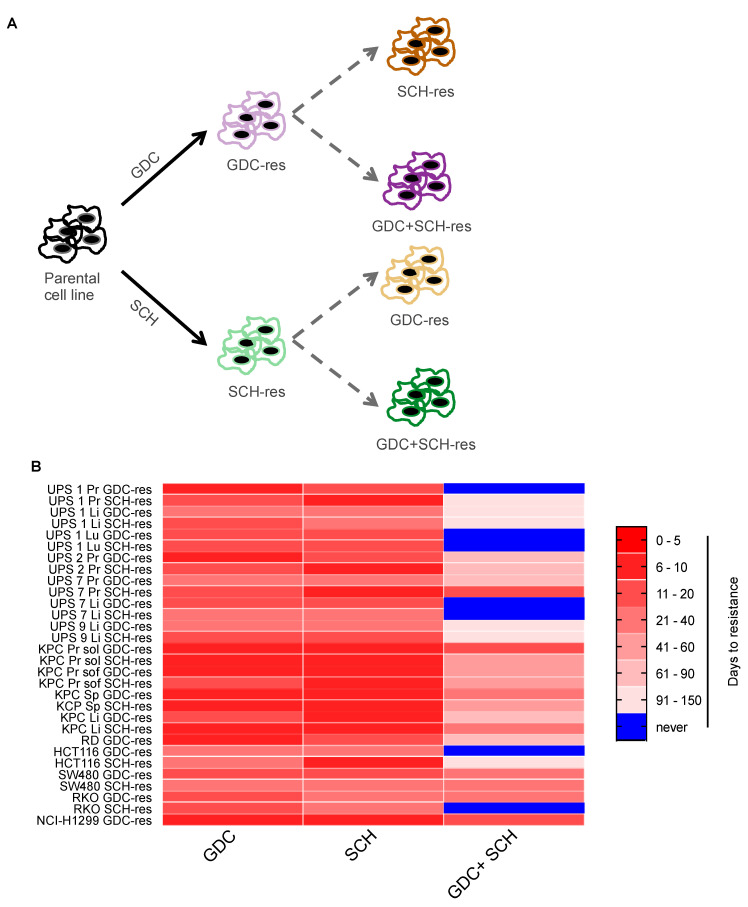
Cells resistant to MEKi or ERKi are frequently resistant to double inhibition. (**A**). Depiction of workflow used to generate secondary resistant cell lines. (**B**). Long-term drug resistance assay. The indicated sub-confluent cell lines (5 × 10^5^ cells/6 cm plate) were cultivated with the indicated inhibitors (1 μM). Days to resistance represents the timepoint at which cells were first split and continued proliferation thereafter, cells that did not survive the treatments or proliferate within 4 months are depicted in blue.

**Figure 7 cancers-13-01852-f007:**
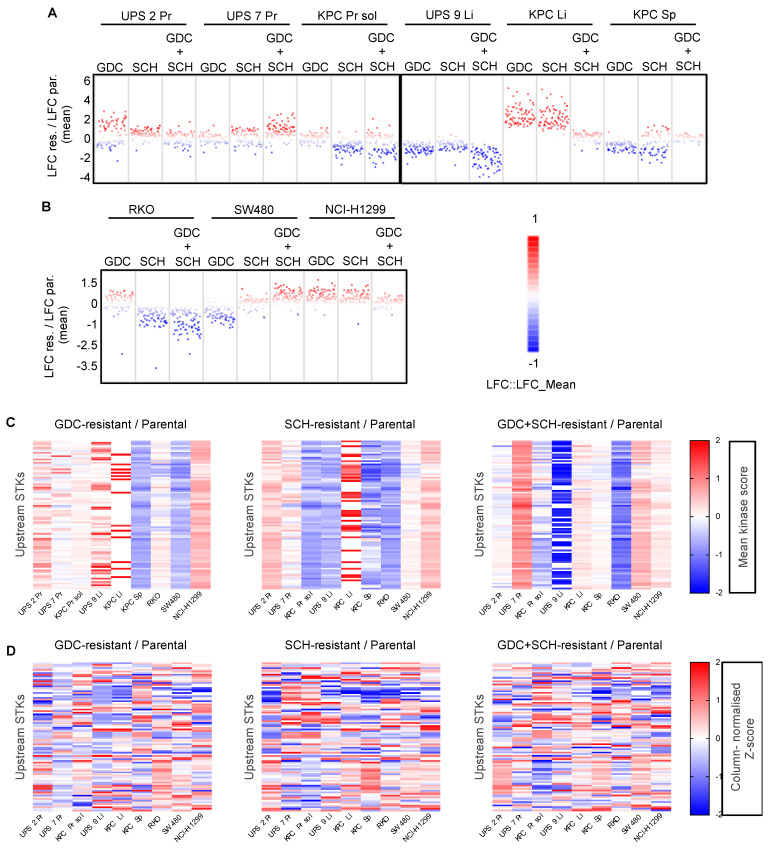
Patterns of STK activity in resistant cells. (**A**,**B**). Log_2_ Fold Change (LFC) distribution of upstream STK kinase activity scores in resistant cell lines relative to their parental untreated counterpart in mouse (**A**) and human (**B**) cell lines. (**C**). Heatmaps depicting ratios of upstream kinase activity scores of resistant relative to parental cells organized by kinases on the Y-axis. (**D**). Heatmaps of column-normalized Z-scores based on upstream kinase activity scores of resistant relative to parental cells organized by kinases on the Y-axis. The Z-score acts as an intra-sample normalization to highlight relative differences in the activity of individual kinases compared to all kinases within the sample.

**Figure 8 cancers-13-01852-f008:**
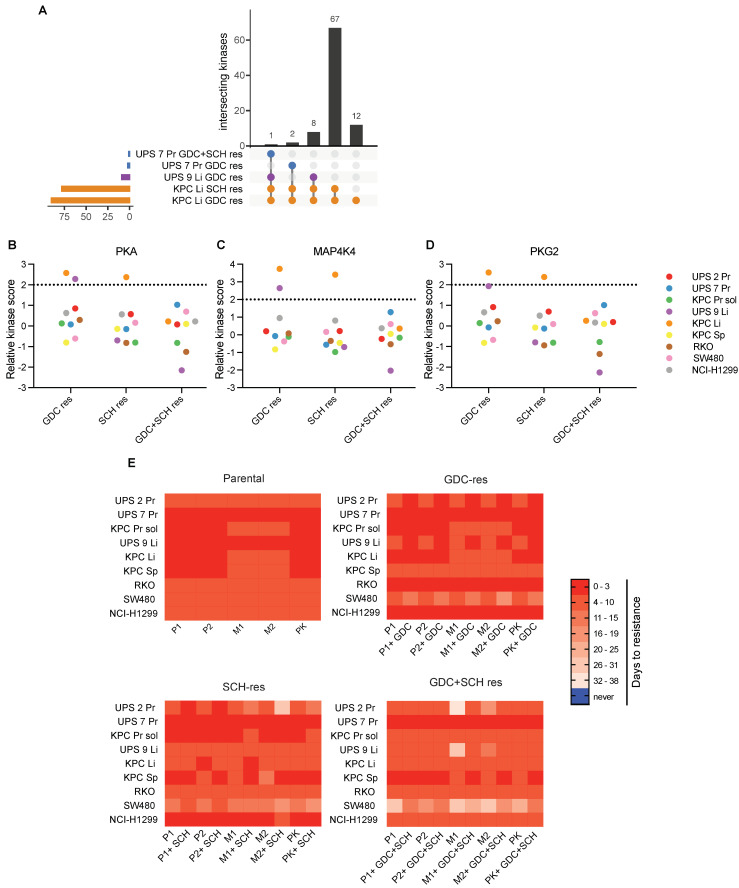
Testing anti-proliferative effects of inhibition of candidate kinases in MEKi and/or ERKi resistant cells. (**A**). UpSetR plot of the overlapping upregulated kinases with LFC mean kinase score greater than or equal to 1.5. The bar plot (top) shows the number of intersecting upregulated kinases among the sets of cell lines that are indicated in the matrix below. The bar plots on the left show the total number of upregulated kinases in each cell line. (**B**–**D**). Ratios of upstream kinase activity scores for PKA (**B**), MAP4K4 (**C**) and PKG2 (**D**) in the indicated resistant cell lines relative to parental cells. (**E**). Long-term drug resistance assay. The indicated sub-confluent cell lines were cultivated with the indicated inhibitors (1 μM). Days to resistance represents the timepoint at which cells were first split and continued to proliferate.

## Data Availability

All data relevant to this manuscript is presented in the figures and supplementary materials.

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
