# Peer review of "Sensitivity and Resistance of Oncogenic RAS-Driven Tumors to Dual MEK and ERK Inhibition"

_cancers, 2021, doi:10.3390/cancers13081852_

Round 1

Reviewer 1 Report

The manuscript by Catalano et al presents a detailed study on the sensitivity and resistance to various drug inhibitors of oncogenic RAS tumors and cancer cells. The authors first present the development of mouse sarcoma models harboring constitutively active Hras or Kras GTPase thanks to lentiviral infection. Then, they show that these sarcoma models, in addition to other cancerous cellular models, are sensitive to dual application of MEK and ERK inhibitors. The combined effect is more potent than single treatment. They also show that the dual treatment prevents the occurrence of resistance, even though the treatment appeared to be highly toxic in vivo. Importantly, they show that if one of the inhibitors is applied first, the emergence of resistance to this inhibitor prevents susceptibility to the dual treatment. Thus the authors suggest that the dual treatment should be always performed in first place, not after treatment with one inhibitor. The emergence of resistance is studied by kinome profiling; showing no clear pattern and suggesting a high heterogeneity of mechanisms. Yet, the authors show that sensitivity to given drug can be restored for each specific acquired resistance.

This is a pretty bulky work, with a lot of different assays and in vivo/in vitro models. Yet, the manuscript reads well and some important insights are given, especially regarding the dual ERK and MEK inhibition and the way it should be done (without a first round of treatment). Unfortunately for further therapies, the dual treatment was highly toxic in the in vivo mouse model but there is some room for optimization regarding the amount and timing of drug application. I don’t have any major revision to suggest and I agree on the publication. Two minor comments:

  • In the title I would remove the word “Mechanisms” since no specific mechanism is found for the emergence of resistance. The title would be sufficient with only “Sensitivity and resistance of oncogenic RAS-driven tumours to dual MEK and ERK inhibition”
  • The quality of the figures can be improved (smaller size for some of the colored matrices, better fonts, use of bold, avoid pixelization for Figure 8)              

Author Response

We are most grateful for the very positive comments of the reviewer.

  • In the title I would remove the word “Mechanisms” since no specific mechanism is found for the emergence of resistance. The title would be sufficient with only “Sensitivity and resistance of oncogenic RAS-driven tumours to dual MEK and ERK inhibition”

Thank you, we agree with this suggestion and have made this change.

  • The quality of the figures can be improved (smaller size for some of the colored matrices, better fonts, use of bold, avoid pixelization for Figure 8)
  • Thank you. This seems to have been partly a problem with the formatting when copying our original figures into the Cancers Word template. We have corrected the pixelation issue. We have not altered the use of bold as I believe that this is not necessary and we have used uniform formatting throughout all figures for consistency. Figures are provided as pasted in PDF so each figure can be easily resized as required by the editor. We welcome any further specific requests for modification of figure formatting by the editorial team should they be necessary  

Reviewer 2 Report

Catalano et al. used a wide range of techniques to study the responses of Ras-driven cancers to RAF, MEK, and ERK kinase inhibitors and their combinations. The study findings illustrate the extreme complexity of molecular mechanisms contributing to cancer cell resistance to small-molecule ERK pathway inhibitors. The results are exciting, the manuscript is well written, and the storyline is clear. This reviewer truly enjoyed reading the manuscript and did not find any significant issues requiring further clarification.

Nevertheless, there are several minor issues that the authors should address before the manuscript is published:
1) There seem to be a problem concerning text formatting. For example, ten to the power of nine is presented in the provided pdf as 109.
2) In Figure 3c-f, text labeling along the bottom side of the heatmaps does not seem correctly formatted. The legend on the right side of the figure lacks color coding.
3) Figure 8 has low resolution in the provided pdf document.
4) A word is missing on line 1058.
5) Lines 204-206: It does not seem correct to cite an article (Ref 55) that does not describe the derivation/generation of the cell lines, especially if they are commonly used and commercially available.

Author Response

We are most grateful for the very positive comments of the reviewer and for the helpful suggestions that we have responded to in the new manuscript as described below.

Nevertheless, there are several minor issues that the authors should address before the manuscript is published:
1) There seem to be a problem concerning text formatting. For example, ten to the power of nine is presented in the provided pdf as 109.

Thank you for noticing this. This problem arose when copying our original document into the Cancers Word template.

2) In Figure 3c-f, text labeling along the bottom side of the heatmaps does not seem correctly formatted. The legend on the right side of the figure lacks color coding.

These issues seem to have been due to the conversion of the file that the reviewer saw as they are not present in the version that we submit. Please advise if these issues persist and we can find a solution.

3) Figure 8 has low resolution in the provided pdf document.

This was also a copy-paste issue that we have now corrected.

4) A word is missing on line 1058.

This text has been corrected.

5) Lines 204-206: It does not seem correct to cite an article (Ref 55) that does not describe the derivation/generation of the cell lines, especially if they are commonly used and commercially available.

Thank you, we have corrected this issue